

# Ice nucleation activity of silicates and aluminosilicates in pure water and aqueous solutions. Part 3 – Aluminosilicates

Anand Kumar, Claudia Marcolli, Thomas Peter

Institute for Atmospheric and Climate Sciences, ETH Zurich, Zurich, 8092, Switzerland

*Correspondence to:* Anand Kumar (anand.kumar@env.ethz.ch)

**Abstract.** Aluminosilicates (such as feldspars, clay minerals and micas) and quartz (a crystalline form of silica), constitute the majority of airborne mineral dust. Despite similarities in structures and surfaces they differ greatly in terms of their ice nucleation (IN) efficiency. Here, we show that determining factors for their IN activity include surface ion exchange, $NH_3$ or $NH_4^+$ adsorption, as well as surface degradation due to the slow dissolution of the minerals. We performed immersion freezing

experiments with the (Na-Ca)-feldspar andesine, the K-feldspar sanidine, the clay mineral kaolinite, the micas muscovite and biotite, and gibbsite ($Al(OH)_3$, a mineral form of aluminum hydroxide) and compare their IN efficiencies with those of the previously characterized K-feldspar microcline and quartz. Samples were suspended in pure water as well as in aqueous solutions of $NH_3$, $(NH_4)_2SO_4$, $NH_4Cl$ and $Na_2SO_4$, with solute concentrations corresponding to water activities $a_w = 0.88 - 1.0$. Using differential scanning calorimetry (DSC) on emulsified micron-sized droplets, we derived onset temperatures of

heterogeneous ($T_{het}$) and homogeneous ($T_{hom}$) freezing as well as heterogeneously frozen water volume fractions ($F_{het}$). Suspensions in pure water of andesine, sanidine and kaolinite yield $T_{het}$ = 242.8 K, 241.2 K and 240.3 K, respectively, while no discernable heterogeneous freezing signal is present in case of the micas or gibbsite (i.e. $T_{het} \approx T_{hom} \approx 237.0$ K). The presence of $NH_3$ and/or $NH_4^+$-salts as solutes has distinct effects on the IN efficiency of most of the investigated minerals. When feldspars and kaolinite are suspended in very dilute solutions of $NH_3$ or $NH_4^+$-salts (< 0.1 molal, corresponding to $a_w \gtrsim 0.99$), $T_{het}$ shifts to

higher temperatures (by 2.6 – 7.0 K compared to the pure water suspension). Even micas and gibbsite develop weak heterogeneous freezing activities in ammonia solutions. Conversely, suspensions containing $Na_2SO_4$ cause $T_{het}$ of feldspars to clearly fall below the water-activity-based immersion freezing description ($\Delta a_w$ = const) even in very dilute $Na_2SO_4$ solutions, while $T_{het}$ of kaolinite follows the $\Delta a_w$ = const curve. The water activity determines how the freezing temperature is affected by solute concentration alone, i.e. if the surface properties of the ice nucleating particles are not affected by the solute. Therefore,

the complex behavior of the IN activities can only be explained in terms of solute-surface-specific processes. We suggest that the immediate exchange of the native cations ($K^+/Na^+/Ca^{2+}$) with protons, when feldspars are immersed in water, is a prerequisite for their high IN efficiency. On the other hand, excess cations from dissolved alkali salts prevent surface protonation, thus explaining the decreased IN activity in such solutions. In kaolinite, the lack of exchangeable cations in the crystal lattice explains why the IN activity is insensitive to the presence of alkali salts ($\Delta a_w$ = const). We hypothesize that adsorption of $NH_3$ and $NH_4^+$

on the feldspar surface rather than ion exchange is the main reason for the anomalous increased $T_{het}$ in dilute solutions of $NH_3$ or $NH_4^+$-salts. This is supported by the response of kaolinite to $NH_3$ or $NH_4^+$, despite lacking exchangeable ions. Finally, on longer timescales (hours to days) the dissolution of the feldspars in water or solutions becomes an important process leading to depletion of Al and formation of an amorphous layer enriched in Si within less than an hour. This hampers the IN activity of andesine the most, followed by sanidine, then eventually microcline, the least soluble feldspar.



## 1 Introduction

Clouds interacting with incoming solar radiation and outgoing longwave radiation influence the Earth's radiation budget (IPCC, 2013). Ice crystals are found in cold cirrus clouds as well as mixed-phase clouds, modulating their radiative properties. Ice can either form homogeneously from liquid droplets supercooled to temperatures below about 237 K; or heterogeneously at higher temperatures with the help of foreign particles, called ice nucleating particles (INPs), which lower the free energy required to form a critical ice embryo growing into crystalline ice. Heterogeneous ice nucleation (IN) can proceed via various pathways, namely (1) immersion freezing, when ice forms on an INP suspended inside a supercooled droplet; (2) condensation freezing, when IN is concurrent with the activation of an aerosol particle to a cloud droplet; (3) contact nucleation, when ice forms in a supercooled droplet upon collision with an INP (Pruppacher and Klett, 1994; Vali et al., 2015). These freezing mechanisms all occur under participation of a liquid phase. Ice may also form at conditions supersaturated with respect to ice but subsaturated with respect to liquid water, a process termed deposition nucleation. Traditionally, it has been ascribed to deposition of water molecules directly from the vapor phase onto a solid INP without involvement of a liquid phase, though it has been questioned whether this process actually occurs, or whether ice forms by pore condensation and freezing (PCF), after stabilizing water in pores by an inverse Kelvin effect (Marcolli, 2014).

Various studies have shown that mineral dust, typically composed of feldspars, clay minerals, quartz, micas, calcite and metal oxides, constitutes an important class of INPs (Pruppacher and Klett, 1994; Murray et al., 2011; Hoose and Möhler, 2012; Atkinson et al., 2013; Cziczo et al., 2013; Kanji et al., 2017). Feldspars have been reported to be the most IN active minerals although the individual members of the feldspar group exhibit very different IN efficiencies (Atkinson et al., 2013; Zolles et al., 2015; Harrison et al., 2016; Kaufmann et al., 2016). During long range transport in the atmosphere, mineral dust particles can acquire organic and inorganic coatings (Usher et al., 2003; Sullivan et al., 2007), which may change their IN efficiencies (Zuberi et al., 2002; Zobrist et al., 2008; Eastwood et al., 2009; Augustin-Bauditz et al., 2014; 2016; Kumar et al., 2018a). When the INPs are fully covered by a water-soluble coating, IN may occur by an immersion freezing mechanism, typically after the aerosol particles experience increasing humidity.

Koop et al. (2000) provided observational evidence and theoretical underpinning that the homogeneous freezing temperatures of aqueous solution droplets can be described as a function of the water activity of solution ($a_w$) alone, namely by shifting the ice melting curve by a constant $\Delta a_w$ to higher $a_w$-values. A similar approach has been used, with a suitably reduced $\Delta a_w$, to describe the immersion freezing temperatures of various types of INPs as a function of $a_w$. While the water-activity-based description of homogeneous IN is well established, for heterogeneous IN this description assumes implicitly that there is no interaction between particle surface and solute. Several studies suggest that the IN efficiency of various INPs can indeed be approximated by such a water-activity-based description (Archuleta et al., 2005; Zobrist et al., 2006; Zobrist et al., 2008; Koop and Zobrist, 2009; Knopf et al., 2011; Knopf and Forrester, 2011; Knopf and Alpert, 2013; Rigg et al., 2013). However, recent studies by Whale et al. (2018) and Kumar et al. (2018a) showed independently that the IN temperatures of some minerals – including the K-feldspars microcline and sanidine – deviate significantly from a freezing point line with $\Delta a_w = $ const, with a shift to higher temperatures in the presence of dilute $NH_3$- and $NH_4^+$-containing solutions and a shift to lower temperatures in the presence of alkali salts.

In Kumar et al. (2018a) we discussed the interactions of inorganic solutes with the microcline surface and the effects on its IN efficiency. In this and the companion paper (Part 2; Kumar et al., 2018b) we relate IN on mineral surfaces with the mineral surface properties by investigating the differences in IN activity of chemically and/or structurally similar minerals in pure water



and aqueous solutions. The analysis of our freezing experiments suggests that heterogeneous IN does not occur on the whole surface of INPs with a uniform probability. Rather, there are preferred locations, so-called active sites, which are responsible for the IN activity of a surface (Fletcher, 1969; Marcolli et al., 2007; Vali, 2014; Vali et al., 2015; Kaufmann et al., 2017). Estimates based on classical nucleation theory suggests that minimum surface areas of these sites need to cover about $10 - 50$ nm$^2$ (Kaufmann et al., 2017). In the present paper, for the discussion of IN activity of aluminosilicates we will assume that IN active surface structures need to be of such size. We will further assume that IN active sites exhibit surface functional groups that are characteristic of the mineral, but either occurring in a special 2D-arrangement (e.g. higher density of certain end-groups on a flat surface) or with a special 3D-feature (e.g. a step, crack or wedge), making them IN active.

In our companion papers (Kumar et al., 2018a; 2018b) we focus on the specific aluminosilicate microcline and on various aluminum-free silicas (both amorphous and crystalline forms) with special focus on quartz (a crystalline form of silica), respectively. The present study investigates the differences in IN activities of a number of other aluminosilicates, and offers an overall summary. The other aluminosilicates investigated by immersion freezing include sanidine (K-feldspar), andesine ((Na-Ca)-feldspar), kaolinite, micas (muscovite and biotite) and gibbsite, dispersed in solution droplets containing ammonia or the inorganic salts ammonium sulfate, ammonium chloride and sodium sulfate.

## 2 Methodology

### 2.1 Mineralogy and size distribution

### 2.2.1 Feldspars

Feldspars are crystalline aluminosilicates, primarily of igneous origin, with the general formula $XAl_{1-2}Si_{3-2}O_8$, often written as $XT_4O_8$ with T = (Al, Si) in tetrahedral coordination with oxygen. X represents an alkali or alkaline earth metal, acting as a charge compensating cation. The tetrahedra with Al at the center carry single negative charges which are compensated by $K^+/Na^+$ (for 1 Al atom) or $Ca^{2+}$ (for 2 Al atoms). Microcline, sanidine and orthoclase (all K-rich feldspars) are polymorphs and form the potassium feldspar group. All of them exhibit an Si:Al ratio of 3:1. In sanidine, which forms at high temperatures, Al and Si are fully disordered with Al taking random positions in the aluminosilicate framework leading to monoclinic symmetry. In microcline, Al is fully ordered leading to triclinic symmetry while orthoclase takes an intermediate position with Al and Si being partially disordered. Despite the difference in charge, $Na^+$ and $Ca^{2+}$ are sufficiently similar in ionic radius, so that there is a complete solid solution series, the plagioclase series, between albite ($NaAlSi_3O_8$) and anorthite ($CaAl_2Si_2O_8$), with intermediate members divided arbitrarily on the basis of anorthite contribution (see Fig. 1). In this series, the Ca:Na and Si:Al ratios are linked with the Al content which changes according to the Ca content. Andesine takes an intermediate position of Na-Ca feldspars with a Ca/(Ca + Na) = $30 - 50$ % (Greenwood and Earnshaw, 1998).

When suspended in water/solutions, feldspars undergo slow dissolution, which is a function of several factors such as crystal surface composition, solution pH, type of ions in solution and exposure time. In Section 4.1.2 we discuss how dissolution is relevant for our experiments and can in fact explain the observed IN ability of feldspars.

### 2.2.2 Kaolinite

Kaolinite, $(Al_4[Si_4O_{10}](OH)_8)$, a layered 1:1 (one tetrahedral sheet and one octahedral sheet forming stacked T-O layers) dioctahedral phyllosilicate, is one of the most abundant clay minerals found in the Earth's crust. Each layer consists of a sheet of



SiO$_4$ tetrahedra forming six-membered silicate rings connected via common oxygen atoms to a sheet of AlO$_6$ octahedra forming four-membered aluminate rings. Weak hydrogen bonds between Al–OH and Si–O–Si groups provide interlayer attraction preventing ions or molecules from entering the interlayer space and constituting the non-swelling nature of this mineral. Since ionic substitution of Si$^{4+}$ with Al$^{3+}$ is absent in the regular kaolinite lattice, exchangeable ions only occur at defects or at edges.

Cleavage of kaolinite preferentially occurs along the basal plane, resulting on one side in a hydroxylated Al-surface with Al atoms arranged in a hexagonal pattern and on the other side in a siloxane surface with Si–O–Si bridges forming hexagonal rings (Bear, 1965). It is still unclear whether these basal surfaces, edges or defects make kaolinite IN active. We discuss this in more detail in Section 4.2.3.

### 2.2.3 Micas

Muscovite (KAl$_2$(AlSi$_3$O$_{10}$)(OH)$_2$) and biotite (K(Mg,Fe)$_3$AlSi$_3$O$_{10}$(OH)$_2$), both quite soft (on Mohs scale 2 – 3), are 2:1 layer phyllosilicates (one octahedral sheet sandwiched between two tetrahedral sheets, forming stacked T-O-T layers) belonging to the mica group with nearly perfect basal cleavage (Bower et al., 2016; Christenson and Thomson, 2016). The tetrahedral layer consists of Si and Al at a ratio of 3:1. The Al is randomly ordered avoiding Al–O–Al arrangements. The presence of Al in the tetrahedral layer introduces a negative charge which is neutralized by potassium ions acting as the bridge between the T-O-T layers. Muscovite and biotite differ in their octahedral layers, which in the case of muscovite is dioctahedral and occupied by Al$^{3+}$. Biotite is part of a solid solution series within the mica group with trioctahedral layers occupied by Fe$^{2+}$ and Mg$^{2+}$. The end members of the series are pure iron biotite called annite and pure magnesium biotite called phlogopite (Bray et al., 2015), while biotites take intermediate positions in terms of Fe:Mg ratios (Fleet et al., 2003). We use this information to correlate the IN ability of mica to its surface composition (see Section 4.3.1).

### 2.2.4 Gibbsite

Gibbsite is an aluminum hydroxide of the general formula Al(OH)$_3$, that exists as different polymorphs. Particles of this hydroxide are of rare abundance in atmospheric dusts. However, we added gibbsite to this study because the hydroxylated Al-sheets of kaolinite are structurally identical with gibbsite. The basal surface of gibbsite is comparable to the aluminol surface of kaolinite and, therefore, often used as a model surface to elucidate the physicochemical properties of the Al-surface of kaolinite (Liu et al., 2015; Kumar et al., 2016).

### 2.2.5 Sources of samples and particle size distribution

Feldspars (sanidine and andesine) and mica (muscovite and biotite) samples were obtained from the Institute of Geochemistry and Petrology of ETH Zurich and milled with a tungsten carbide ball mill. Particle number size distribution was obtained with a TSI 3080 scanning mobility particle sizer (SMPS) and a TSI 3321 aerodynamic particle sizer (APS). The dry particles were dispersed using a fluidized bed. The detailed size distribution and mineralogical composition of the feldspars, kaolinite and muscovite are given in Kaufmann et al. (2016). Biotite shows a bimodal particle size distribution with mode diameters of 241 nm and 1.7 μm (see Supplementary Information). X-ray diffraction (XRD) analysis showed no traces of tungsten carbide in any of the milled samples. The kaolinite used in this study was KGa-1b from the Clay Mineral Society. Gibbsite (Al(OH)$_3$.xH$_2$O, < 45 μm) was obtained from Sigma Aldrich.

### 2.2 Emulsion freezing experiments with mineral dusts freshly suspended in pure water/solutions



We described the general setup of immersion freezing experiments in Part 1 of this series of papers (Kumar et al. (2018a)). Here we repeat essential aspects for convenience. The experiments were carried out with the DSC (TA Instruments, Q10) setup

(Zobrist et al., 2008). Sanidine and andesine suspensions (2 wt %), kaolinite suspensions (5 wt %), mica (muscovite and biotite; 5 wt % and 10 wt %) suspensions and gibbsite (10 wt %) suspensions in water (molecular biology reagent water from Sigma-Aldrich) were prepared with varying concentrations (0 − 20 wt %) of $(NH_4)_2SO_4$ (Sigma Aldrich, ≥ 99 %), $NH_4Cl$ (Sigma Aldrich, ≥ 99.5 %), $Na_2SO_4$ (Sigma Aldrich, ≥ 99 %), and diluted $NH_3$ solutions (Merck, 25 %). To avoid particle aggregation, the suspensions were sonicated for 5 min before preparing the emulsions. The aqueous suspension and an oil/surfactant mixture

(95 wt % mineral oil (Sigma Aldrich) and 5 wt % lanolin (Fluka Chemical)) taken in a ratio of 1:4 were mixed using a rotor-stator homogenizer (Polytron PT 1300D with a PT-DA 1307/2EC dispersing aggregate) for emulsification (40 seconds at 7000 rpm). 4 − 10 mg of this emulsion was placed in an aluminum pan, which was hermetically closed, and then following the method developed and described by Marcolli et al. (2007) three freezing cycles in the DSC were performed. The first and the third freezing cycles were executed at a cooling rate of 10 K min$^{-1}$ to control the stability of the emulsion. The second freezing cycle

was run at 1 K min$^{-1}$ cooling rate and used for evaluation (Zobrist et al., 2008; Pinti et al., 2012; Kaufmann et al., 2016; Kumar et al., 2018a). Emulsions prepared by this procedure exhibit droplet size distributions peaking at diameters of about 2 − 3 μm in number and a broad distribution in volume with highest contributions from particles with diameters between 4 and 12 μm similar as the ones shown in Figs. 1 of Marcolli et al. (2007), Pinti et al. (2012) and Kaufmann et al. (2016). For a clear heterogeneous signal, dust particles need to be of similar size and smaller than the droplets. Large particles (> 10 μm) present in dust samples

contribute significantly to the dust mass but hardly to the heterogeneous freezing signal.

Typically, a DSC thermogram of the cooling cycle performed with an emulsion containing INPs features two freezing signals, as depicted in Kumar et al. (2018a). The first peak occurring at a warmer temperature displays the heat release accompanied by heterogeneous freezing and the second peak occurring at a colder temperature is due to homogeneous freezing. The freezing temperatures ($T_{het}$ and $T_{hom}$) are determined as the onset of the freezing peak (i.e., intersection of the tangent drawn at the point of

greatest slope at the leading edge of the thermal peak with the extrapolated baseline). Droplets with diameters of about 12 μm are considered to be relevant for the freezing onset. The loading of these droplets with particles depends on the particle size distribution and the suspension concentration. For 2 − 5 wt % suspensions, 12 μm droplets contain about 100 − 1000 particles. The melting temperature ($T_{melt}$) was determined as the maximum of the ice melting peak. For the investigated samples, average precision in $T_{het}$ were ± 0.1 K with maximum deviations not exceeding 0.5 K. $T_{hom}$ and $T_{melt}$ are precise within ± 0.1 K.

The heat release is related to the frozen water volume and is given by the integral of the heat signal over time. Since the enthalpy of freezing is temperature dependent, this evaluation is only approximate (Speedy, 1987; Johari et al., 1994). $F_{het}$ is defined as the ratio of the heterogeneous freezing signal to the total freezing signal (heterogeneous and homogeneous). More details about the evaluation of $T_{het}$ and $F_{het}$ can be found in Kumar et al. (2018a). Absolute uncertainties in $F_{het}$, are on average ± 0.02 and do not exceed ± 0.1 in cases where the heterogeneous freezing signal is clearly distinguishable from the homogeneous freezing

signal. It is important to highlight that $F_{het}$ carries larger uncertainties (> ± 0.1) in cases where heterogeneous freezing signals are weak and overlap (forming a flattened shoulder; see Supplementary Material) with the homogeneous freezing signal (e.g. in case of strong hampering of IN ability of feldspars in alkali solutes; see Section 3.1.2). Spikes occurring before the appearance of the heterogeneous freezing signal are excluded from the evaluation as they originate from single droplets (mostly between 100 − 300 μm with some up to 500 μm in diameter) in the tail of the droplet size distribution, which are orders of magnitude larger in

volume than the average droplets, and not representative for the sample.





For feldspars, freezing experiments were at least performed in duplicates with separate emulsions, prepared from a single suspension for each concentration and the means are reported. For kaolinite and micas, freezing experiments were performed with emulsions prepared from at least two separate suspensions for each solute concentration and means are reported. Representative DSC thermograms of all experiments are shown in the Supplementary Material.

**2.3 Aging experiments with kaolinite suspended in pure water/solutions**

Similar to our experiments with microcline (Kumar et al., 2018a), we let kaolinite (5 wt % suspension) age in pure water, ammonia solution (0.005 molal), and ammonium sulfate solutions (0.1 wt % and 10 wt %) over a period of five days and its IN activity was tested during this period in emulsion freezing experiments with the DSC setup. For each solute concentration two separate suspensions were prepared and aged. Small portions were taken from the suspension and emulsified for freezing experiments (as described in the previous section) on the day of suspension preparation (fresh) and the subsequent five days in order to assess the long-term effect of ammonia and ammonium containing solutes on the IN efficiency of kaolinite.

**3 Results**

**3.1 Feldspars**

**3.1.1 Dependence of the heterogeneous freezing temperatures on water activity**

Figures 2a and 2b show mean heterogeneous ($T_{het}$) and homogeneous freezing onsets ($T_{hom}$) and ice melting temperatures ($T_{melt}$) of the feldspars sanidine and andesine for all investigated solutes ($NH_4Cl$, $(NH_4)_2SO_4$, $Na_2SO_4$) as a function of water activity of solution ($a_w$). The water activity is defined as the ratio of the equilibrium vapor pressure of water over the flat surface of the solution and the saturation vapor pressure over the flat surface of pure water at the same temperature. We derive $a_w$ from the melting point depression measured by DSC during the heating cycle (thus, the measured melting points, $T_{melt}$, lie by definition exactly on the melting curve). This procedure was not applicable to $Na_2SO_4$, because above the eutectic concentration of 4.6 wt% a hydrate of $Na_2SO_4$ crystallizes together with ice. Therefore, water activities for $Na_2SO_4$ solutions have been calculated based on the solute concentration using the AIOMFAC thermodynamic model at 298 K (Zuend et al., 2008; 2011). The homogeneous freezing curve (dotted black line) is obtained by a constant shift of the melting curve by $\Delta a_w^{hom}(T) = 0.294$, calculated to best fit the current dataset (see Kumar et al. (2018a) for more details of the derivation). This offset is in good agreement with $\Delta a_w^{hom}(T) = 0.305$ found by Koop et al. (2000). Following Koop et al. (2000), we assume $a_w$ to be temperature independent between $T_{hom}$ and $T_{melt}$.

A constant offset $\Delta a_w^{het}$ is also applied to the heterogeneous freezing temperatures. Here, the offset in $a_w$ is chosen so that the heterogeneous freezing line passes through the freezing temperature of the pure water case. This yields the solid black lines with $\Delta a_w^{het} = 0.264$ for sanidine and $\Delta a_w^{het} = 0.254$ for andesine, which will be referred to as $T_{het}^{\Delta a_{w,san}}(a_w)$ and $T_{het}^{\Delta a_{w,and}}(a_w)$, respectively. The heterogeneous freezing behavior would be expected to follow these curves, if specific chemical interactions between the solute and the ice-nucleating surface were absent, so that the only effect of the solute is a freezing point depression. However, as can be seen from Figs. 2a and 2b, the measured heterogeneous freezing onset temperatures, $T_{het}$, deviate from the $T_{het}^{\Delta a_w}(a_w)$-curves for both feldspars. For both $NH_4^+$-solute cases ($(NH_4)_2SO_4$ and $NH_4Cl$), sanidine shows a strong increase in $T_{het}$ at low solute concentrations ($a_w > 0.99$). Remarkably, temperature maxima are up to $\approx 7$ K above $T_{het}^{\Delta a_{w,san}}(a_w)$. This increase is



followed by a decrease at higher concentrations back to $T_{\text{het}}^{\Delta a_{w},\text{san}}(a_w)$ in case of NH$_4$Cl solutions ($a_w \approx 0.925$) and even slightly below $T_{\text{het}}^{\Delta a_{w},\text{san}}(a_w)$ in case of (NH$_4$)$_2$SO$_4$ solutions ($a_w \leq 0.98$). Andesine also shows an increase in $T_{\text{het}}$, though smaller in magnitude ($\approx 2.6$ K with respect to the pure water case), but this enhancement persists to higher NH$_4^+$ concentrations. The observed enhancements with respect to $T_{\text{het}}^{\Delta a_{w},\text{and}}(a_w)$ are $\approx 5.1$ K and $\approx 4.0$ K for NH$_4$Cl and (NH$_4$)$_2$SO$_4$, respectively.

In contrast to NH$_3$/NH$_4^+$-solutions, freezing experiments in the presence of Na$_2$SO$_4$ as a non-NH$_4^+$ solute show a strong decrease in $T_{\text{het}}$ below $T_{\text{het}}^{\Delta a_{w}}(a_w)$ for both feldspars even at the lowest investigated solute concentrations (0.0005 wt %, $a_w > 0.99$).

### 3.1.2 Dependence of the heterogeneously frozen fractions on water activity

Figure 2c and 2d show the heterogeneously frozen fraction $F_{\text{het}}$ (the ratio of the heterogeneous freezing signal to the total freezing signal) as a function of $a_w$ for sanidine and andesine. In dilute NH$_4^+$-containing solutions, sanidine shows a remarkable enhancement in $F_{\text{het}}$ when compared to the suspension in pure water. With decreasing water activities, this enhancement reverses into a decline (for $a_w < 0.99$ in case of (NH$_4$)$_2$SO$_4$ and for $a_w < 0.98$ in case of NH$_4$Cl). In contrast, $F_{\text{het}}$ of andesine is less influenced by the presence of NH$_4^+$-solutes: there is no significant enhancement at low concentrations and much less of a decrease at higher concentrations.

For both feldspars, the addition of Na$_2$SO$_4$ leads to a strong decrease of $F_{\text{het}}$ for the lowest solute concentration ($a_w \geq 0.99$) and even to an almost complete inhibition of the IN activity for $a_w < 0.99$.

## 3.2 Kaolinite

### 3.2.1 Dependence of the heterogeneous freezing temperatures on water activity

Figure 3a presents $T_{\text{het}}$, $T_{\text{hom}}$ and $T_{\text{melt}}$ as a function of $a_w$ for kaolinite. The offset in $a_w$ applied to shift the melting curve so that it passes through the freezing temperature of the pure water case for kaolinite is $\Delta a_w^{\text{het}} = 0.272$, here referred to as $T_{\text{het}}^{\Delta a_{w},\text{kaol}}(a_w)$. In the presence of NH$_4^+$-solutes, kaolinite shows an increase ($\approx 2.4$ K) in $T_{\text{het}}$ compared to $T_{\text{het}}^{\Delta a_{w},\text{kaol}}(a_w)$ even at the lowest investigated solute concentration (0.1 wt %; $a_w > 0.99$). The influence of NH$_3$ on the IN activity was found to be similar to the one of the NH$_4^+$-containing solutes. This enhancement relative to $T_{\text{het}}^{\Delta a_{w},\text{kaol}}(a_w)$ persists over the complete concentration range probed in this study, similar to the case of andesine. For $a_w < 0.9$, kaolinite shows an increase up to 2.9 K, 5.5 K and 6.9 K in $T_{\text{het}}$ compared to $T_{\text{het}}^{\Delta a_{w},kaol}(a_w)$ in solutions of (NH$_4$)$_2$SO$_4$, NH$_4$Cl and NH$_3$, respectively. In contrast to the feldspars, in Na$_2$SO$_4$ solutions $T_{\text{het}}$ follows $T_{\text{het}}^{\Delta a_{w},kaol}(a_w)$ within the error range for all investigated concentrations.

### 3.2.2 Dependence of the heterogeneous frozen water volume fractions on water activity

Figure 3b shows $F_{\text{het}}$ as a function of $a_w$ for kaolinite. Interestingly, kaolinite shows a strong enhancement in $F_{\text{het}}$ compared to the pure water case in the presence of NH$_3$ and NH$_4^+$-solutes over the complete concentration range. This enhancement is strongest for ammonia solutions, reaching $F_{\text{het}} \approx 1$ ($F_{\text{het}} \approx 1$ was assumed when it was not possible to discern a homogeneous freezing signal; see Fig. 7) at $a_w = 0.98 - 0.96$, compared to $F_{\text{het}} \approx 0.5$ in pure water. In contrast, $F_{\text{het}}$ seems to be unaffected by Na$_2$SO$_4$ within experimental uncertainties over the measured concentration range.





### 3.3 Heterogeneous freezing of aqueous solution droplets containing micas

Extending our investigation of the effect of $NH_3$- and $NH_4^+$-containing solutes on the IN activity of aluminosilicates, we performed emulsion freezing experiments also with mica dust particles suspended in solution droplets. Figure 4 shows DSC thermograms of muscovite particles suspended in $(NH_4)_2SO_4$ and $NH_3$ solution droplets of increasing concentration, and Fig. 5

the same for biotite in $NH_3$ solution droplets. $T_{het}$ and $F_{het}$ from these measurements for muscovite and biotite are summarized in Tables 1 and 2, respectively. Interestingly, when suspended in pure water, neither muscovite nor biotite exhibit a discernible heterogeneous freezing signal in the DSC thermograms. However, both start to develop IN activity when immersed in ammonia or ammonium solutions (indicated by the dashed black lines in Figs. 4 and 5). The frozen fraction, $F_{het}$, is a strong function of $NH_3$- or $NH_4^+$-concentration. For muscovite (5 wt %), $F_{het}$ increases from 0.138 in 0.005 molal $NH_3$ ($a_w = 0.997$) to 0.387 in 4.5

molal $NH_3$ ($a_w = 0.922$) solutions. In the case of biotite (5 wt %), suspensions do not reveal a heterogeneous freezing signal in pure water or in dilute ammonia solutions. A well discernable heterogeneous freezing signal appears only in concentrated ammonia (4.5 molal, $a_w = 0.922$, Figure 5). For 10 wt % biotite suspensions, the heterogeneous freezing signal becomes visible already in 2 molal ammonia ($a_w = 0.968$; Table 2), yet the signals are clearly weaker than those from muscovite suspensions at similar solute concentrations.

### 3.4 Heterogeneous freezing of aqueous solution droplets containing gibbsite

Figure 6 shows the DSC thermograms of the emulsion freezing experiments carried out with 10 wt % gibbsite suspensions in pure water, in $NH_3$- (panel a) and in $(NH_4)_2SO_4$-containing solutions (panel b). Gibbsite, exhibiting no heterogeneous freezing signal during emulsion freezing experiments in pure water, starts to show a weak heterogeneous freezing signal with increasing $NH_3$ concentration (indicated by the dashed black line in Fig. 6a). On the other hand, no heterogeneous freezing signal can be

observed for gibbsite suspended in $(NH_4)_2SO_4$ solutions (0.05 – 10 wt %; $a_w = 0.996 – 0.961$).

### 4. Discussion

#### 4.1 Feldspars

$T_{het}$ and $F_{het}$ trends for sanidine and andesine are similar to the ones of the K-feldspar microcline reported in our previous study, Kumar et al. (2018a), however, with significant variations. All investigated feldspars show an increase in $T_{het}$ at low

concentrations of $NH_4^+$-solutes ($a_w \geq 0.99$). This increase is highest for sanidine and lowest for andesine. On the other hand, the decrease of $T_{het}$ at higher solute concentration is most pronounced for microcline with values falling below the prediction from the water activity-based description. For andesine, $T_{het}$ remains above $T_{het}^{\Delta a_{w,and}}(a_w)$ while it just decreases to $T_{het}^{\Delta a_{w,san}}(a_w)$ for sanidine. In case of non-$NH_4^+$-solutes, $T_{het}$ decreases below the prediction from the water activity-based approach for all investigated feldspars even at low solute concentrations.

Similar trends are also observed for $F_{het}$. Microcline shows an increase in $F_{het}$ at low concentration of $NH_4^+$-solutes, but also a strong decrease at high solute concentrations. Sanidine shows a similar behavior as microcline, but the decrease of $F_{het}$ at high $NH_4^+$-solute concentrations is less pronounced. On the other hand, $F_{het}$ of andesine is strongly decreased only in the presence of $Na_2SO_4$ but hardly affected by the presence of $NH_4^+$-solutes. These findings are also in general agreement with droplet freezing experiments by Whale et al. (2018) who observed for microcline and sanidine an increase of active site densities in the presence

of $NH_4^+$-solutes and a decrease in alkali halide solutions.




The differences in $T_{het}$ and $F_{het}$ in pure water and the investigated solutions indicate substantial differences between the investigated feldspar surfaces, despite their similar crystal structures. Microcline and sanidine are both K-feldspars with Si:Al ratios of about 3:1 but with Al and Si ordered in the microcline crystal lattice and disordered in sanidine. Andesine is a (Na-Ca)-feldspar with Ca:(Na+Ca) = 30 – 50 % and the Si:Al ratio varying accordingly. In the following, we will discuss how these differences between the feldspars influence their surface structure in the presence of water and solutes, and how the surface structure can be related to the measured $T_{het}$ and $F_{het}$.

### 4.1.1 Surface ion exchange

In pure water, the native charge-balancing surface cations ($K^+/Na^+/Ca^{2+}$) immediately undergo cation-exchange by $H^+/H_3O^+$ (Chardon et al., 2006; Lee et al., 2008), with hardly any simultaneous structural surface damage (Busenberg and Clemency, 1976). The native cation may also participate in ion-exchange with an externally added cation depending on the size and charge compatibility of the latter with the crystal structure (Auerbach et al., 2003; Belchinskaya et al., 2013). Ammonium ions not only have a strong preference for cation exchange with K-feldspars and (Na-Ca)-feldspars but are also fixed to the surface in non-exchangeable form, because of the high bonding energy involved (Nash and Marshall, 1957; Barker, 1964; Russell, 1965; Chou and Wollast, 1989; Dontsova et al., 2005).

In Kumar et al. (2018a), we have shown that the ion exchange is unlikely as a reason for the enhanced IN efficiency of microcline in dilute $NH_4^+$-containing solutions. Rather, hydrogen bonding of ammonia/ammonium with the surface hydroxyl groups seems to provide better ice-like template sites for the incoming water molecules (Anim-Danso et al., 2016), leading to an increase in $T_{het}$.

### 4.1.2 Aluminum depletion and surface dissolution

In addition to the exchange of surface ions discussed above (and in more detail in Kumar et al. (2018a)), feldspars undergo slow dissolution in water and aqueous solutions followed and accompanied by the precipitation of more stable phases like kaolinite, gibbsite or halloysite (Stillings and Brantley, 1995). Depending on the specific feldspar and solution pH, steady-state Si dissolution rates vary between $10^{-10}$ and $10^{-14}$ moles m$^{-2}$ s$^{-1}$ (Crundwell, 2015) with lowest values at near neutral conditions. The solution pH during the emulsion freezing experiments reported in Fig. 2 are close to neutral with pH values ranging from 5.5 – 6.7. For the feldspars investigated in this study, steady-state dissolution rates are lowest for microcline with $4\cdot10^{-14} - 8\cdot10^{-14}$ moles m$^{-2}$ s$^{-1}$ at pH ~ 6 (Crundwell, 2015) followed by sanidine with a rate of ~ $2\cdot10^{-13}$ moles m$^{-2}$ s$^{-1}$ at near neutral conditions (Crundwell, 2015), while dissolution of andesine occurs with a steady-state rate of $10^{-12} - 10^{-11}$ moles m$^{-2}$ s$^{-1}$ at pH ~ 8 (Gudbrandsson et al., 2014). Hereby, initial dissolution rates of freshly suspended feldspar may be higher by up to three orders of magnitude than the rates at steady-state (Zhu et al., 2016). With these rates, dissolution is in the range of one monolayer within the timescale of a DSC freezing experiment (1 – 1.5 h) for andesine, and below it for sanidine and microcline.

At near-neutral conditions, the dissolution proceeds via protonation of the oxygen of ≡Al–O–Si≡ bridges with subsequent release of $Al^{3+}$ resulting in an incongruent (deviation of ratio of released Si to Al from stoichiometric ratio) initial dissolution for most feldspars when they are freshly suspended in water (Oelkers and Schott, 1995; Oelkers et al., 2009), and leading to a ≡Si–OH rich surface (Oxburgh et al., 1994; Stillings and Brantley, 1995; Oelkers et al., 2009). Oelkers et al. (2009) found aluminum surface depletion to occur readily in their 20-min titration experiments with the Na-feldspar albite. Thus, the feldspar surfaces in pure water suspensions can be considered at least partly or even completely depleted in aluminum with the dangling bonds



replaced by silanol groups within the timescales of our experiments. The dissolution incongruence with respect to Al-atoms depends on the Si/Al ordering and the Si:Al ratio of the feldspar lattice (Yang et al., 2014a; 2014b). Yang et al. (2014b) investigated the stoichiometry of feldspar dissolution under acidic conditions (pH 1.8) and found that microcline with Si:Al = 3.0

dissolved with a ratio of Si:Al = 2.1. Their sanidine sample with Si:Al = 2.87 released Al with Si:Al = 1.36, while their andesine sample with Si:Al = 1.95 dissolved at a ratio Si:Al = 0.76:1. This dissolution incongruence leads to a layer leached in Al, with a thickness that depends on the specific feldspar and increases with decreasing pH. At steady-state, this Al-depleted layer is 6.6 – 8.5 nm thick for microcline at pH 1 (Lee et al., 2008), but only 1 – 2 nm at pH 3 (Stillings and Brantley, 1995). For andesine, it reaches 60 – 120 nm at pH 3.5 and still 15 – 30 nm at pH 5.7 (Muir et al., 1990). Since Al dissolution at steady-state conditions

needs to occur through this layer, the andesine surface is considered to consist of somewhat loosened and distorted feldspar chains resulting in a porous, amorphous-like structure (Marshall, 1962; Stillings and Brantley, 1995; Zhang and Lüttge, 2007). At pH 1, the first 5 nm of surface layer of microcline were found to be amorphous (Lee et al., 2008). Since amorphous silica shows barely any IN activity (see also the companion paper Kumar et al. (2018b)), the loss of IN activity of microcline in the presence of $H_2SO_4$ (Kumar et al., 2018a), may be explained by the presence of an amorphous silica layer. Since $T_{het}$ correlates

with the thickness of the leached layer reported for the investigated feldspars, we hypothesize that a thick amorphous surface layer hampers the IN activity of feldspars.

### 4.1.3 Surface charge and surface protonation

Feldspars exhibit a negative surface charge over a wide pH range with a point of zero charge (PZC) < 2 (Karagüzel et al., 2005; Vidyadhar and Hanumantha Rao, 2007). Bringing the feldspar surface closer to the PZC by adding $H_2SO_4$ did not increase $T_{het}$

but decreases $F_{het}$ due to surface degradation as was shown for microcline (Kumar et al., 2018a). This shows that surface charge is only one among several factors influencing IN activity. We discuss the effect of surface charge on IN activity of mineral surfaces in more detail in Sect. 5.

Surface hydroxyl groups are considered to promote IN because they can form hydrogen bonds with water molecules and bring them in a suitable orientation for IN. In a recent molecular simulation study, Pedevilla et al. (2017) found that the OH density

rather than a specific OH pattern is a useful descriptor of IN ability. The relevance of surface hydroxylation is in agreement with our observations that $Na^+$ ions added to an aqueous feldspar suspension decrease $T_{het}$ and $F_{het}$ for all investigated feldspars since they replace the protons (Oelkers et al., 2009) thus decreasing the surface hydroxylation (see Fig. 2 of this paper; Kumar et al. (2018a)). Ammonium on the other hand, can form hydrogen bonds with water molecules and does therefore not reduce the IN activity. In addition, $NH_3/NH_4^+$ is not only involved in ion exchange but also binds to the feldspar surface (Kumar et al., 2018a).

Therefore, it increases the capacity for hydrogen bonding even more and leads to an increased IN activity. A strong increase of proton concentration (low pH), on the other hand, hampers or even totally impedes IN because it promotes aluminum depletion and the formation of an amorphous silica surface layer.

### 4.1.4 Influence of sulfate

While we consider solute effects to be dominated by cations at low concentrations, the anions seem to become more relevant at

higher concentrations. Aqueous phase complexes of dissolved aluminum with sulfate enhance the solubility of aluminum. Surface complexes of sulfate with surface aluminum can impede or enhance dissolution of aluminum. Bidentate surface complexes with sulfate do not enhance surface protonation and should stabilize aluminum at the surface, while the formation of mononuclear complexes should facilitate surface protonation and help dissolution of aluminum (Min et al., 2015). The effect of




sulfate should therefore depend on the bonding strength of aluminum to the feldspar surface, which depends on the Si/Al order
and Si:Al ratio. Min et al. (2015) found that the dissolution rate of anorthite, a (Na-Ca)-feldspar with Si:Al = 1, is enhanced by
the formation of monodentate complexes between the aluminum surface sites and sulfate. This seems to be also the case for the
(Na-Ca)-feldspar andesine. On the other hand, surface complexation of aluminum sites with sulfate seems to decrease the
removal of aluminum and block sites for IN in the case of microcline, while sanidine takes an intermediate position.

### 4.1.5 Previous IN studies with feldspars

The IN activity of feldspars has already been investigated in previous studies. All studies agree that the IN activity of K-feldspars
are high among mineral dusts (Atkinson et al., 2013; Harrison et al., 2016; Kaufmann et al., 2016; Peckhaus et al., 2016).
Kaufmann et al. (2016) found that the IN activity of microcline is superior compared with other K-feldspars. Several studies
have been dedicated to microcline. Niedermeier et al. (2015) and Burkert-Kohn et al. (2017) found frozen fractions of 0.5 at 244
– 245.5 K for condensation freezing on 300 nm microcline particles that were strongly reduced when the samples were aged
under acidic conditions. A strong reduction of IN activity when feldspars were coated with sulfuric acid was also observed by
Kulkarni et al. (2014). The IN activity of microcline after aging in pure water seems to depend on the specific sample. Samples
with the highest onset freezing temperatures also show the strongest reduction in IN activity after aging, indicating that the best
active sites might also be the most labile ones (Harrison et al., 2016; Peckhaus et al., 2016). Harrison et al. (2016) found that one
sample of Na-feldspar (albite) was similarly active as microcline when freshly suspended in water, but lost its high activity after
having been suspended over months in water, indicating that its IN activity may be related to sites with high solubility that are
lost over time by dissolution. This confirms that protonation conveys on one hand sites for IN but may destabilize on the other
hand active sites. Kiselev et al. (2016) found that high-energy (100) surface patches of K-rich feldspars are best suited for
deposition growth of aligned ice crystals below water saturation. However, it is not clear how resistant these high-energy sites
are when immersed in water. Whale et al. (2017) related the exceptionally high IN ability of K-feldspars to microtextures, giving
rise to topographic features with high IN temperatures. The IN site densities that they investigated with their setup were in the
range from $10^{-1}$ to $10^{3}$ cm$^{-2}$ with IN temperatures up to 271 K. With the DSC emulsion freezing experiments, we investigate the
properties of sites that are common to submicrometer particles, i.e. with surface densities in the range from $10^{-10}$ to $10^{-5}$ cm$^{-2}$
nucleating ice up to 252 K. While the sites investigated by Whale et al. (2017) may be well correlated with microtextures, it is
unlikely that such features are common enough in submicrometer particles to account for the IN activity observed in our
emulsion freezing experiments.

### 4.2 Kaolinite

### 4.2.1 IN efficiency in pure water

Kaolinite has shown IN activity in various freezing modes (Zuberi et al., 2002; Welti et al., 2009; Lüönd et al., 2010; Murray et
al., 2011; Hoose and Möhler, 2012; Pinti et al., 2012; Nagare et al., 2016). Many studies investigated K-SA kaolinite from Sigma
Aldrich (Zuberi et al., 2002; Lüönd et al., 2010; Burkert-Kohn et al., 2017), while others used the KGa-1b kaolinite from the
Clay Mineral Society. Pinti et al. (2012) found with their emulsion freezing experiments that K-SA exhibits a second freezing
peak at higher temperatures with onset at ~248 K that could be related to impurities of feldspars (Atkinson et al., 2013). On the
other hand, KGa-1b features only one heterogeneous freezing peak with an almost constant onset at ~240 K in emulsion freezing
experiments irrespective of the kaolinite concentration in the aqueous suspension (Pinti et al., 2012), which is an indication of the
homogeneous composition and mineralogical purity of the sample. The $T_{het}$ for 5 wt% kaolinite suspension observed in our study



is in agreement within measurement uncertainties with Pinti et al. (2012). Kaufmann et al. (2016) evaluated the active particle fraction of kaolinite KGa-1b to be only about 0.04 compared to values of 0.54 – 0.64 determined for microcline. A lower active fraction of kaolinite compared with the feldspar microcline was also confirmed by single particle immersion freezing experiments (Burkert-Kohn et al., 2017). With $T_{het}$ = 240 K, the onset of heterogeneous IN is lower than the ones of microcline

($T_{het}$ = 252 K), sanidine ($T_{het}$ = 241 K), and andesine ($T_{het}$ = 243 K).

### 4.2.2 IN efficiency in solutions

The DSC curves for kaolinite suspended in $NH_3$ solution (Fig. 7) show that for a certain $NH_3$ concentration range (0.5 – 2 molal; $a_w$ = 0.988 – 0.955) the heterogeneous freezing signal totally dominates the homogeneous one, indicating that initially IN inactive particles (contributing to the homogeneous freezing DSC signal) became IN active after addition of ammonia. Salam et

al. (2007; 2008) have previously reported improved IN efficiency of the clay mineral montmorillonite below and above water saturation after the particles were exposed to $NH_3$ gas.

Kaolinite shows an increase of $T_{het}$ and $F_{het}$ in dilute $NH_4^+$-solutes and dilute $NH_3$ solutions but no decrease at higher concentrations, similar to the behavior of andesine but opposite to the one of microcline. One notable difference between the investigated feldspars and kaolinite is that the latter shows no decline of IN activity in the presence of $Na_2SO_4$.

The similarities in the effect of $NH_3/NH_4^+$-solutes on the IN activity of kaolinite and the investigated feldspars indicate similarities of the chemical functionalization of IN active sites in these two groups of minerals. In the following, we compare the surface properties of kaolinite with the ones of the feldspars to elucidate which surface properties can explain the similarities and discrepancies in IN activity.

### 4.2.3 Which kaolinite surface is responsible for the IN activity?

As a sheet aluminosilicate, kaolinite exhibits two basal faces – the siloxane tetrahedral and the alumina octahedral – and hydroxylated edges. The overall surface charge of kaolinite in pure water is negative with a PZC < 2. The PZC increases when kaolinite is suspended in salt solutions (Yukselen-Aksoy and Kaya, 2011). However, this overall value determined from bulk experiments does not need to apply to the individual faces of kaolinite.

Recent surface force measurements have been able to probe the surface properties of the kaolinite faces individually. These

measurements evidenced a negative charge of the siloxane surface with a PZC below pH 4 (Gupta and Miller, 2010). Isomorphic substitution of Si(IV) for Al(III) in tetrahedral layers is considered the main reason for the net permanent negative surface charge, which is balanced by the adsorption of cations (e.g., $Na^+$, $K^+$, $Ca^{2+}$) (Cashen, 1959; Grim, 1968; Bolland et al., 1980). The oxygen atoms on the siloxane surface are relatively weak electron donors and not capable to form hydrogen bonds with water molecules. Therefore, the near-surface water molecules interact predominantly with each other so that the surface is

considered nearly hydrophobic (Jepson, 1984; Giese and van Oss, 1993; Braggs et al., 1994).

Unlike the siloxane surface, the hydroxyl-rich surface of the Al-octahedral layer is hydrophilic and forms hydrogen bonds with water (Schoonheydt and Johnston, 2006; Yin et al., 2012). Therefore, the two basal surfaces show very different surface chemistry (Tunega et al., 2002; Tunega et al., 2004). The alumina face was found to be negatively charged at pH ≥ 8 and positively at pH ≤ 6 indicating a PZC of pH 6 – 8 (Gupta and Miller, 2010). The pH-dependent, non-permanent surface charge



arises from protonation/deprotonation of surface hydroxyl groups. While silanol groups mainly contribute to negative charge through the formation of $SiO^-$ by deprotonation, aluminol groups undergo both protonation at low pH and deprotonation at high pH (Abendroth, 1970; Bleam et al., 1993).

Disruption of the kaolinite sheets leads to dangling oxygen atoms at the crystal edges which takes up water molecules to transform into hydroxyl groups. This process produces a surface rich in silanols and aluminols, which terminate tetrahedral and
octahedral layers, respectively (Brady et al., 1996; Liu et al., 2013). The PZC of the kaolinite edge surface was found to be below pH 4. The negative charge is considered to arise from deprotonation of Si-sites which is only partly counteracted by positively charged Al-sites (Brady et al., 1996). The reactive hydroxyl groups, due to their charge, have the potential to chemisorb certain other ionic species (Schoonheydt and Johnston, 2006, 2013). Such hydroxyl groups are also present at steps and cracks.

First principles calculations and molecular dynamics simulations have indicated that hydrogen bonding to surface hydroxyl
groups is an essential factor for providing IN activity to mineral surfaces (Hu and Michaelides, 2007; Sosso et al., 2016a; Glatz and Sarupria, 2018). The absence of sites for hydrogen bonding make the regular siloxane surface an unlikely candidate for IN (Freedman, 2015). The alumina surface, on the other hand, is OH-rich and has been probed in several modeling studies for its capability to nucleate ice. Based on a density-functional theory study, Hu and Michaelides (2007) concluded that the basal surfaces of kaolinite do not support epitaxial multilayer ice growth, rather they may promote the growth of the prism face of ice
(Cox et al., 2013). Monte Carlo simulations carried out by Croteau et al. (2008, 2009) suggested that a rigid Al-surface of kaolinite (001) is incapable of orienting water molecules into ice-like configurations, instead trenches and surface defects were suggested to be responsible for IN (Croteau et al., 2010a, b). Zielke et al. (2016) showed that both the Al-surface and the Si-surface can nucleate ice, by reorientation of hydroxyl groups in the former case and by ordered arrangement of hexagonal and cubic ice layers joined at their basal planes in the latter case. Glatz and Sarupria (2018), based on molecular dynamic simulations
on kaolinite-like surfaces, argued that lattice matching and hydrogen bonding are necessary but not sufficient conditions for IN. Sosso et al. (2016b) found that IN on the hydroxylated basal surface of kaolinite proceeds exclusively via the formation of the hexagonal ice polytype but that this process crucially depends on very small structural changes upon kaolinite surface relaxation in the molecular dynamics simulations.

While most of these studies focused on the basal planes, experimental evidence suggests that the edges are preferred locations for
IN. Indeed, Wang et al. (2016) found in IN experiments with a cell coupled to an environmental scanning electron microscope that ice preferentially nucleates at the edges of kaolinite particles when they were exposed to increasing RH. The similar response of kaolinite and feldspars to $NH_3/NH_4^+$-solutes indicates a similar chemical composition of IN active sites on the surfaces of these two mineral types, suggesting that the kaolinite edges, rich in both, aluminol and silanol groups should be responsible for the observed IN activity.

In order to assess the likeliness of IN occurring at the Al-surface of kaolinite and how it is influenced by the presence of $NH_3/NH_4^+$-solutes, we ran emulsion freezing experiments with gibbsite (10 wt %) in pure water and different concentrations of $NH_3$ and $(NH_4)_2SO_4$ solutions. Similar to the hydroxylated Al-layers of kaolinite, gibbsite consists of stacked sheets of linked octahedra of aluminum hydroxide. Therefore, the gibbsite surface is often taken as a model for the Al-surface of kaolinite (Liu et al., 2015; Kumar et al., 2016). The PZC of gibbsite falls in the pH range from 7.5 to 11.3 (Kosmulski, 2009; Liu et al., 2015),
thus in a similar, yet somewhat higher range than the one of the Al-surface of kaolinite (pH 6 – 8). DSC thermograms of the emulsion freezing experiments with gibbsite (Fig. 6) show no heterogeneous freezing signal in pure water and $(NH_4)_2SO_4$



solutions up to 10 wt %. Only when suspended in $NH_3$ solutions (0.05 − 1 molal, $a_w$ = 0.996 - 0.981), a weak heterogeneous freezing signal developed. This indicates that $NH_3$ hydrogen bonded to the hydroxylated Al-surface provide IN activity to gibbsite, while the positively charged $NH_4^+$ does not adsorb on the positively charged Al-surface. Given the inability of gibbsite

to nucleate ice in water and $(NH_4)_2SO_4$, we conclude that the IN activity of kaolinite rather stems from the edges as suggested by Wang et al. (2016) and Croteau et al. (2010a). This conclusion is further supported by the finding that kaolinite indeed adsorbs $NH_3$ by forming hydrogen bonds with the hydroxyl groups at the edges (James and Harward, 1962).

### 4.2.4 Aging effect

We carried out immersion freezing experiments with kaolinite (5 wt %) suspended in pure water, $NH_3$ solution (0.005 molal, $a_w$
= 0.999), and $(NH_4)_2SO_4$ solutions (0.1 wt % and 10 wt %; $a_w$ = 0.996, $a_w$ = 0.961, respectively) over a period of five days to assess the effect of aging on the IN efficiency of kaolinite. Figure 8 shows $T_{het}$ (panel a) and $F_{het}$ (panel b) as a function of time. $T_{het}$ remains stable over the measured time period within the experimental measurement uncertainties. Similarly, $F_{het}$ remains constant except in the pure water case where it shows a slight increase.

The dissolution rate of kaolinite depends on pH with the lowest values of $10^{-14} − 10^{-13}$ moles (of Si) $m^{-2}$ $s^{-1}$ realized at near
neutral conditions (Carroll and Walther, 1990; Huertas et al., 1999). At acidic conditions the dissolution increases because of $H^+$ attack at the Si–O–Al linkages of the edges leading to the liberation of aluminum ions into the solution (Xiao and Lasaga, 1994; Fitzgerald et al., 1997). At alkaline conditions, dissolution is considered to occur dominantly at the basal octahedral face via deprotonation of aluminum sites (Huertas et al., 1999; Naderi Khorshidi et al., 2018). At near neutral conditions, both mechanisms are inefficient leading to a minimum in the dissolution rate. Between pH 5 and 10, kaolinite dissolution is
accompanied by the precipitation of an aluminum hydroxide phase (Huertas et al., 1999).

The pH of the solutions used for the aging experiments with kaolinite are in the slow dissolution near-neutral regime. Since there is no decrease in IN efficiency during the 5 days of aging in the aqueous solutions of ammonia and ammonium sulfate, surface alteration during this time period seem to be minor. Indeed, with a dissolution rate of $10^{-14} − 10^{-13}$ moles $m^{-2}$ $s^{-1}$, the Si dissolved during the 5 days of the experiments, should be well below a monolayer and should not lead to a significant alteration of the
surface composition. Nevertheless, when dissolution occurs preferentially on chemically more reactive sites that are likewise IN active, destruction of these sites might lead to a decrease in IN activity. However, the increase of IN activity in pure water rather suggests the generation of new IN active sites due to surface changes in the presence of water.

### 4.3 IN efficiency of micas

The mica minerals, muscovite and biotite, do not show a heterogeneous freezing signal in emulsion freezing experiments with pure water (see Figs. 4 and 5) even in 10 wt % suspensions. Considering the particle size distributions of the two minerals, which
peaks at about 335 nm in case of the muscovite sample (Kaufmann et al., 2016) and is bimodal with maxima at 241 nm and 1.7 μm, the lack of a heterogeneous freezing signal cannot be ascribed to the predominance of empty emulsion droplets. When probing the IN activity of muscovite in the presence of $NH_3$, a weak heterogeneous freezing signal appears at $NH_3$ concentration of 0.005 molal ($a_w$ = 0.997) which develops into a clear shoulder at the highest ammonia concentrations (Table 1). While a clear
heterogeneous freezing signal in form of a shoulder appears when muscovite is suspended in dilute $(NH_4)_2SO_4$ solutions (0.05 − 1 wt %, $a_w$ = 0.996 − 0.988). For biotite, a heterogeneous freezing signal becomes visible only at the highest ammonia



concentration of 4.5 molal ($a_w$ = 0.922) for the 5 wt % suspension and between 2 - 5.5 molal ($a_w$ = 0.968 – 0.912) ammonia solutions for the 10 wt % suspension (Table 2).

Micas have been tested for IN activity in numerous studies, however, with very diverse outcome. Older studies have reported
dentritic ice growth on the basal planes of freshly-cleaved micas (muscovite and fluorophlogopite: $KMg_3AlSi_3O_{10}F_2$) when they were exposed to water saturation at cold temperatures (Bryant et al., 1959; Hallett, 1961; Layton and Harris, 1963). Shen et al. (1977) showed that fluorophlogopite and muscovite particles (44 – 74 µm) induce IN in bulk freezing experiments with onset temperatures of 272 K and 268 K, respectively. Steinke (2013) investigated the freezing of water droplets on a muscovite surface in a cold stage and found IN activity around 250 K but with a much lower active site density compared with clay minerals.
Atkinson et al. (2013), on the other hand, observed hardly any IN activity in immersion mode for a non-specified mica. Campbell et al. (2015) found freezing close to the homogeneous freezing temperature for droplets on muscovite. Surface imperfections on the basal plane of muscovite were found to promote IN on muscovite exposed to water vapor below the threshold temperature for homogeneous IN but had no effect when the surface was immersed in water above this threshold temperature. Overall, the IN activity of micas seems to be much lower than that of clay minerals such as kaolinite despite their similar structure.

**4.3.1 IN activity in relation to the surface properties of micas**

Micas are 2:1 phyllosilicates (see Sect. 2.2.3) that are easily cleaved along the basal plane yielding molecularly smooth surfaces. The basal surfaces consist of a tetrahedral layer with an Si:Al ratio of 3:1. This surface is dominated by Si–O–Si and Al–O–Si bridges exhibiting little hydroxylation. Due to the isomorphic substitution of Si(IV) for Al(III) in the tetrahedral layer, the basal surfaces are hydrophilic and carry a permanent negative charge that is neutralized by $K^+$ ions (Zhao et al., 2008; Yan et al.,
2013). When suspended in electrolyte solutions, $K^+$ exposed to the surface undergo ion exchange within seconds (Cho and Komarneni, 2009; de Poel et al., 2017; Lee et al., 2017). In pure water surface $K^+$ and to a lesser degree, also interlayer potassium ions are lost to the suspension and replaced by protons introducing hydroxylation to the basal surface and widening the muscovite lamellae (Banfield and Eggleton, 1990). At the edges of the plate-like particles, the broken bonds of the disrupted sheets are saturated by –OH, resulting in a hydrophilic, hydroxylated surface with a surface charge that depends on pH (Zhao et
al., 2008).

Micas slowly dissolve when suspended in water accompanied by precipitation of metal (hydr)oxides and kaolinite, which may form nanometer coatings on the mica surface blocking active sites (Pachana et al., 2012). The dissolution occurs with similar pH dependence as observed for feldspars with the lowest rates at near neutral conditions (Oelkers et al., 2008; Bray et al., 2015; Lammers et al., 2017). Dissolution may occur via etch pits on the basal surface, but the dominating process seems to be
corrosion of the edge surfaces (Oelkers et al., 2008; Pachana et al., 2012).

The Si dissolution rate of muscovite is about $10^{-13}$ to $10^{-12}$ moles $m^{-2} s^{-1}$ at neutral conditions (Brady and Walther, 1989; Lammers et al., 2017), which is sufficient to lead to surface alterations within the timescale of our experiments. The initial dissolution of Al is higher than the one of Si (Pachana et al., 2012), confirming that edges dissolve more readily than the basal faces. The basal plane of muscovite carries negative charge independent of pH while the edges carry a negative charge at high pH and a positive
charge at low pH with a PZC at pH 7 – 8. The dissolution rate of biotite is $10^{-12}$ to $10^{-11}$ moles $m^{-2} s^{-1}$ at neutral conditions. The initial dissolution at pH 6 was observed to be congruent with respect to Si and Al, but Mg and Fe positioned in the octahedral layer dissolved at a higher rate, resulting in a metal depleted disordered octahedral layer (Pachana et al., 2012; Bray et al., 2015).

Atmos. Chem. Phys. Discuss.



The absence of a heterogeneous freezing signal in emulsion freezing experiments suggests a low density of IN active sites. Thus, the IN activity seems to stem from very special features that are rare, while the regular mica surfaces are inactive despite the hydrophilicity of the basal surfaces and the dense hydroxylation of the edges, both characteristics of IN active surfaces. Indeed, sum-frequency-generation (SFG) spectra showed that water adsorbed at full monolayer coverage (90 % RH) forms an ice-like film on the basal muscovite surface (Miranda et al., 1998). Yet, this film does not seem to grow readily into bulk ice. The surface electric field of the negatively charged muscovite surface orders water molecules with the protons pointing towards the surface. SFG spectra show that this ordering decreases when the water freezes and increases again when the ice melts, suggesting that the ordered water layer adsorbed on the basal mica surface is unable to template ice (Anim-Danso et al., 2016). This finding is in-line with Abdelmonem et al. (2017) who found that IN was not promoted by the presence of an ordered water layer on the hydroxylated sapphire ($\alpha$-Al$_2$O$_3$) surface. On the contrary, IN was observed to occur at slightly warmer temperature close to the PZC when the water molecules were disordered. While at low and high pH, the aligning of the water molecules on the positively or negatively charged surface, respectively, was detrimental to IN.

While the absence of IN activity at the edges of biotite may be ascribed to the fast disintegration of this surface in the presence of water, it is not clear whether surface alteration at the edges of muscovite are sufficient to explain their lack of IN activity in pure water. In the presence of NH$_3$ and (NH$_4$)$_2$SO$_4$ solutions, both investigated micas show IN activity in the emulsion freezing experiments. This might be due to adsorption of NH$_3$/NH$_4^+$ on the basal planes increasing the number of sites available for hydrogen bonding or due to adsorption at the edges which may influence the dissolution rate.

## 5 Summary of IN activity of mineral surfaces

### 5.1 Influence of surface properties

Table 3 summarizes the IN activity of the minerals investigated in this study together with results of microcline from Part 1 (Kumar et al., 2018a) and quartz results from Part 2 (Kumar et al., 2018b) and relates them to selected surface properties of the investigated minerals, namely the prevalent surface groups, the dissolution rates at near neutral conditions and the PZC.

Isomorphic substitution of Si(IV) for Al(III) in aluminosilicates imparts them a net permanent negative surface charge. In addition, non-permanent, pH-dependent surface charge may arise from protonation/deprotonation of surface hydroxyl groups. Although all investigated minerals possess hydroxylated surfaces, which are generally thought to promote IN (Hu and Michaelides, 2007; Sosso et al., 2016a; Glatz and Sarupria, 2018) (see Table 3), not all of them proved to be IN active in our emulsion freezing experiments. Namely, the gibbsite and mica (muscovite and biotite) samples did not show any heterogeneous freezing signal in pure water. Zeta potential measurements of these minerals show that the hydroxylated edges of muscovite and the surface of gibbsite carry a positive charge at neutral conditions due to protonation of aluminol groups (see Table 3). This is in contrast to the surfaces of quartz, feldspars and the hydroxylated edges of kaolinite, which all carry negative charge at neutral conditions due to partial deprotonation of the silanol groups. Thus, positively charged (due to protonation) hydroxylated surfaces have a tendency to be IN inactive, while negatively charged surfaces tend to be IN active. However, long-term aged quartz surfaces (7 months; see Kumar et al. (2018b)) show hardly any IN activity while milled quartz surfaces are highly IN active, although the quartz surface is in both cases negatively charged and highly hydroxylated (Turci et al., 2016). On the other hand, AgI (Edwards and Evans, 1962; Marcolli et al., 2016) and sapphire (Abdelmonem et al., 2017) show the highest IN activity close to the point of zero charge. Conversely, pyroelectric LiTaO$_3$ and SrTiO$_3$ promote IN when their surfaces are positively charged





(Ehre et al., 2010). This shows that the OH-surface density together with the surface charge are not sufficient to predict IN
activity.

5.2 Dissolution and growth in pure water

Our emulsion freezing experiments revealed IN activity of all investigated feldspars when suspended in pure water albeit with different $T_{het}$ and $F_{het}$. The investigated feldspars carry a negative surface charge and exhibit similar chemical compositions although with different Si:Al ratios and Si/Al ordering, which both influence the dissolution rate: measurements show that low
Si/Al order and a low Si:Al ratio both increase the dissolution rate. The initial dissolution of most feldspars is incongruent with preferential dissolution of aluminum, resulting in a Si-rich amorphous layer. We hypothesize that the surface remains well-ordered over a longer period of time for microcline due to its lowest dissolution rate (compared to the other investigated feldspars). Indeed, our emulsion freezing experiments on a microcline suspension (2 wt%) in pure water which was aged for 6 months showed no significant change in IN efficiency compared with a freshly prepared sample (aged sample: $T_{het}$ = 251.4 K,
$F_{het}$ = 0.71, see Fig. S15, Supplementary Material; fresh sample: $T_{het}$ = 252.1 K, $F_{het}$ = 0.74 (Kumar et al., 2018a)). Peckhaus et al. (2016) observed a decrease in IN activity by 2 K for a K-feldspar over a time span of 5 months. This implies that the high initial IN activity of microcline degrades only slowly within days or months (Kumar et al., 2018a). The IN activity of the sanidine and andesine samples may be lower than the one of microcline because their surface degradation is relevant already during the first DSC freezing cycles due to their higher dissolution rates. Hydrogen bonding to the feldspar surface is a
prerequisite to arrange water molecules in a suitable configuration for IN, but at the same time, it is the first step to the disintegration of the feldspar surface. Feldspar dissolution occurs via protonation of Al–O–Si bridges followed by removal of $Al^{3+}$ and protonation of the dangling bonds resulting in a silanol-rich surface. As the dissolution proceeds, the Al-depleted Si-layer becomes thicker and more amorphous-like.

The IN ability of quartz and amorphous silica is highly variable from IN inactive to very active. When the surface of milled
quartz grows or dissolves, active sites are lost and the IN activity decreases (Kumar et al., 2018b). Under growth conditions, i.e. when the dissolved Si concentration exceeds the saturation concentration with respect to quartz, a siliceous layer forms on the quartz surface within about a day, that hampers the IN activity of quartz. When this layer is washed away with pure water after several days of aging, the IN activity is restored. On timescales of months, quartz slowly grows within this layer resulting in an intact, grown quartz surface that is barely IN active. This suggests that the IN activity of silica mainly stems from activation
through grinding and is absent in regular, grown quartz surfaces. We ascribe the absence of IN on the highly hydroxylated grown quartz surface to networks of interconnected hydrogen bonds that are too strong to be disrupted by water molecules. This implies that for a hydroxylated surface to be IN active the hydrogen bonds must be available for bonding to water molecules in order to direct them into an ice-like pattern.

5.3 Influence of solutes

Figure 9 reviews the influence of solutes on the heterogeneous IN onset temperatures of the investigated aluminosilicates and quartz. The presence of $NH_3/NH_4^+$ in the suspensions increases $T_{het}$ of all investigated feldspars and kaolinite (Fig. 9, point 1). Moreover, it provides IN activity to the investigated micas and gibbsite, which showed no IN activity in our emulsion freezing experiments performed in pure water (Fig. 9, point 4). The increase in $T_{het}$ of kaolinite with no exchangeable cations confirms the findings from Kumar et al. (2018a), that the enhanced IN activity in the presence of $NH_3/NH_4^+$ is not due to ion exchange of $NH_4^+$
with the native cations, but stems from the adsorption of $NH_3/NH_4^+$ on the mineral surface. Water molecules can form hydrogen





bonds with NH groups of adsorbed $NH_3/NH_4^+$ that may promote their arrangement into an ice-like pattern. Moreover, this adsorption can decrease the feldspar dissolution by stabilizing the surface. While $NH_3/NH_4^+$ also interact with silica surfaces, these interactions do not lead to an increase in $T_{het}$. Rather, a decrease is observed in the presence of $NH_3$ due to the enhanced dissolution of quartz under alkaline conditions (Fig. 9, point 5).

The alkali salt $Na_2SO_4$ strongly decreases the IN activity of feldspars (Fig. 9, point 2) but does not influence that of kaolinite (Fig. 9, point 3). This indicates that alkali ions influence the IN activity of feldspars through ion exchange, which does not take place in kaolinite. Dissolved $Na^+$ may take the positions of the protons on the surface and thus impede IN. This shows that the replacement of the charge balancing cations with protons taking place in pure water is crucial for the IN activity of feldspars.

5.4 Influence of milling

Milling increases the IN activity of quartz (Fig. 9, point 6), while the growth of an intact quartz layer on top of the milled quartz surface strongly hampers it (Fig. 9, point 7). The influence of milling on the IN efficiency of various mineral surfaces has recently garnered attention. Milling leads to an increase in surface irregularities and defects which may lead to changes in the abundance and distribution of surface functional groups. These changes, in general, enhance IN efficiency but the enhancement depends on the type of mineral surface. Zolles et al. (2015) showed that additional milling of K-feldspar leads to only a slight
increase in its high IN efficiency. Kaufmann et al. (2016) performed emulsion freezing experiments with a dust sample which was ground collected in Antarctica (48 % feldspars, 24 % quartz) and observed an increase of the already high IN activity of the sieved sample when it was additionally milled. We ascribe the high IN activity of the sieved sample to the presence of the feldspars, whereas an additional increase after milling is most likely due to activation of the quartz component. Together with the finding that K-feldspars aged in pure water over months (see Sect. 5.2) kept their IN activity, we conclude that mechanical
activation, e.g. due to milling, is not a prerequisite for the IN activity of feldspars. This is in contrast to our findings for quartz surfaces presented here (Fig. 9) and in Kumar et al. (2018b). Boose et al. (2016) found a slightly higher active site density of the milled fraction of their Atacama dust sample (65 % feldspars, 10 % quartz) compared with the sieved fraction (51 % feldspars, 16 % quartz). On the other hand, they observed a decreased IN activity of their Israel dust sample after milling (80 % calcite, 6 % quartz) compared with the sieved fraction (67 % calcite, 7 % quartz). A natural quartz sample from Zolles et al. (2015)
showed a very strong enhancement in IN efficiency due to milling. Moreover, milled samples from Boose et al. (2016) collected in Australia and Morocco, which primarily consisted of quartz showed a very high IN activity, corroborating our results on quartz and amorphous silica in the companion paper (Kumar et al. (2018b)). Therefore, it seems likely that milling is a prerequisite for the IN activity of quartz. A natural process, closest to milling, would be dry erosion, yet, it is unclear whether the forces exerted by fracturing during erosion are sufficient to generate similar defects as milling.

**6 Conclusions and atmospheric implications**

Immersion freezing experiments with aluminosilicates, namely feldspars, kaolinite, micas and gibbsite presented in this study and in Kumar et al. (2018b) have shown that modifications of the mineral surfaces in the presence of water and solutes influence their IN activity.

The interaction of water with mineral surfaces is complex and manifold. It depends not only on the type of mineral but also on
the exposure time of the surfaces to water and solutes. Factors influencing the IN ability of aluminosilicate surfaces are: adsorption and ion exchange with solute molecules influencing the density of OH and NH groups which provide sites for



hydrogen bonding with water molecules, permanent and pH-dependent surface charge influencing the orientation of water molecules and surface changes generated by the slow dissolution of the minerals. Therefore, for an improved understanding of IN, the specific surface properties of each mineral and the processes modifying the mineral surfaces in water and aqueous

solutions need to be investigated by experimental surface techniques. In addition, molecular dynamics simulations of IN may contribute to clarifying these processes, but only if interactions of the mineral surface with water e.g. surface ion exchange, protonation/deprotonation of the surface and slow dissolution are taken into account.

Aluminosilicates constitute the majority of airborne mineral dust. Dust particles can be exposed to reactive gases like $NH_3$ or $SO_2$ during long-range transport, resulting in a coating and surface modification (Sullivan et al., 2007; Kolb et al., 2010;

Fitzgerald et al., 2015). The modifications introduced by chemical coatings depend on the particle mineralogy, transport pathway and exposure duration (Matsuki et al., 2005; Sullivan et al., 2007; Rodríguez et al., 2011). Our results suggest that the IN activity of dust aerosols does not only depend on their composition, but also on their chemical exposure history. In Part 1 (Kumar et al., 2018a) we discussed in detail the fate of microcline in atmospheric solution droplets (especially $(NH_4)_2SO_4$ and $H_2SO_4$) following various atmospheric air parcel trajectories with increasing moisture. This discussion is applicable to the

aluminosilicates shown in this study and can hence be extended. A coating of aqueous $(NH_4)_2SO_4$ solution may enhance the IN efficiency in condensation mode of aluminosilicates in general, but only when sulfuric acid and ammonia are deposited concomitantly. Aluminosilicates, especially feldspars, are sensitive to highly acidic conditions due to enhanced dissolution. This might hamper their relevance in case of long-range transport when they are exposed to acidic air masses (e.g. acquiring sulfuric acid coatings prior to neutralization by ammonia). Moreover, in Part 2 (Kumar et al., 2018b) we have shown that the relevance of

quartz particles as atmospheric INP is uncertain. To assess the IN activity of naturally eroded quartz, IN experiments need to be carried out with quartz-rich natural dust samples that are just sieved and not milled.

*Data availability.* The data for freshly prepared sanidine, andesine and kaolinite suspensions in water or aqueous solutions (Figs. 2 - 3) and aging tests on kaolinite (Fig. 8) presented in this publication are available at the following repository https://doi.org/10.3929/ethz-b-000260067.

*Acknowledgements.* This work was supported by the Swiss National Foundation, project number 200020_156251. We thank the following colleagues from ETH Zürich: Annette Röthlisberger and Marion Rothaupt for helping in carrying out BET and XRD measurements and Dr. Michael Plötze for carrying out detailed XRD analysis; Fabian Mahrt for providing the SMPS and the APS for size distribution measurements; Peter Brack for providing various minerals. We also like to thank Dr. Alexei Kiselev from Institute of Meteorology and Climate Research (Karlsruhe Institute of Technology) for providing valuable feedback on this

manuscript.

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



**Table 1** Summary of the freezing experiments with emulsified aqueous solution droplets containing muscovite (5 wt % and 10 wt %). Note that the absolute uncertainty in $F_{het}$ may be up to $\pm$ 0.1. Only $F_{het}$ above this threshold are clear evidence of heterogeneous IN.

| Solute | Solute Concentration ($m^{\#}$/wt%) | $a_w$ | $T_{hom}$ (K) | $T_{het}$ (K) | | $F_{het}$ | |
|---|---|---|---|---|---|---|---|
| | | | | Muscovite 5wt% | Muscovite 10wt% | Muscovite 5wt% | Muscovite 10wt% |
| $NH_3$ | 4.5/5.5*m | 0.922/0.912* | 225.4/223.6* | 233.4 | 230.6* | 0.387 | 0.471* |
| $NH_3$ | 2m | 0.958 | 232.1 | 236.1 | 237.2 | 0.231 | 0.415 |
| $NH_3$ | 1m | 0.976 | 234 | 238.2 | 238.6 | 0.172 | 0.265 |
| $NH_3$ | 0.5m | 0.987 | 235.5 | 239.2 | 239.9 | 0.143 | 0.386 |
| $NH_3$ | 0.05m | 0.996 | 236.5 | 240.7 | 240.3 | 0.116 | 0.225 |
| $NH_3$ | 0.005m | 0.997 | 236.8 | 240.3 | 239.4 | 0.138 | 0.109 |
| $NH_3$ | 0.0005m | 0.999 | 237.0 | n/a | n/a | n/a | n/a |
| $(NH_4)_2SO_4$ | 1wt% | 0.988 | 236.0 | 240.8 | 243.1 | 0.221 | 0.169 |
| $(NH_4)_2SO_4$ | 0.5wt% | 0.994 | 236.4 | 241.1 | 242.8 | 0.292 | 0.249 |
| $(NH_4)_2SO_4$ | 0.05wt% | 0.996 | 236.9 | 240.5 | 241.8 | 0.199 | 0.415 |
| Pure Water | - | 1 | 236.8 | n/a | n/a | n/a | n/a |

*10wt % muscovite suspended in 5.5 molal $NH_3$ solution; #molality;
n/a: no $T_{het}$ and $F_{het}$ can be reported due to absence of a discernible heterogeneous freezing signal in the emulsion freezing experiments

**Table 2** Summary of the freezing experiments with emulsified aqueous solution droplets containing biotite (5 wt % and 10 wt %). Note that the absolute uncertainty in $F_{het}$ may be up to $\pm$ 0.1. Only $F_{het}$ above this threshold are clear evidence of heterogeneous IN.

| Solute | Solute Concentration (molality) | $a_w$ | $T_{hom}$ (K) | $T_{het}$ (K) | | $F_{het}$ | |
|---|---|---|---|---|---|---|---|
| | | | | Biotite 5wt% | Biotite 10wt% | Biotite 5wt% | Biotite 10wt% |
| $NH_3$ | 4.5/5.5* | 0.922/0.912* | 225.0/223.7* | 228.1 | 232.4* | 0.187 | 0.308* |
| $NH_3$ | 2 | 0.968 | 232.7 | n/a | 235.6 | 0.000 | 0.214 |
| $NH_3$ | 1 | 0.981 | 234.3 | 239.5 | 239.4 | 0.026 | 0.082 |
| $NH_3$ | 0.5 | 0.989 | 235.7 | 240.1 | 240.3 | 0.020 | 0.092 |
| $NH_3$ | 0.05 | 0.998 | 236.7 | 240.0 | 240.2 | 0.042 | 0.063 |
| $NH_3$ | 0.005 | 0.999 | 236.9 | n/a | n/a | n/a | n/a |
| $NH_3$ | 0.0005 | 0.999 | 236.8 | n/a | n/a | n/a | n/a |
| Pure Water | - | 1 | 236.8 | n/a | n/a | n/a | n/a |

*10wt % biotite suspended in 5.5 molal $NH_3$ solution;
n/a: no $T_{het}$ and $F_{het}$ can be reported due to absence of a discernible heterogeneous freezing signal in the emulsion freezing experiments




**Table 3.** IN activities in terms of $T_{het}$ and $F_{het}$ together with surface properties of minerals investigated in this study at near neutral conditions (representative for pure water), microcline from Kumar et al. (2018a) and quartz from Kumar et al. (2018b).

| Mineral | Surface functional groups at neutral conditions | pH of PZC | Dissolution rate (moles Si m$^{-2}$ s$^{-1}$) | Suspension concentration (wt%) | $T_{het}$ (K) pure water | $F_{het}$ pure water |
|---|---|---|---|---|---|---|
| **Feldspars** | Si-O-Si, Si-O-Al, Si-OH-Al, Si-OH, Si-O$^-$, Al-OH, Al-OH$_2^+$ [ref a] | | | | | |
| Microcline (K-feldspar) | | < 2 [ref h] | $4 \cdot 10^{-14} - 8 \cdot 10^{-14}$ at pH ~6 [ref p] | 2 | 252.1 | 0.74 |
| Sanidine (K-feldspar) | | | $\sim 2 \cdot 10^{-13}$ at pH ~6 [ref p] | 2 | 241.2 | 0.42 |
| Andesine (Na/Ca-feldspar) | | < 2 [ref i] | $10^{-12} - 10^{-11}$ at pH ~8 [ref q] | 2 | 242.8 | 0.65 |
| **Kaolinite (clay mineral)** | | | $10^{-14} - 10^{-13}$ at pH ~7 [refs r, s] | 5 | 240.3 | 0.52 |
| *Basal Si face* | Si-O-Si [ref b] | < 4 [refs k, l] | | | | |
| *Basel Al face* | Al-OH$^{1/2-}$, Al-OH$_2^{1/2+}$, Al$_2$-OH [ref c] | 6 – 8 [refs k, l] | | | | |
| *Edges* | Si-O$^-$, Si-OH, Al-OH, Al-O$^-$, Al-OH$_2^+$, Al-(OH)$_2$ [refs c, d] | < 4 [refs k, m] | | | | |
| **Muscovite (mica)** | | | $10^{-13} - 10^{-12}$ at pH ~6 [refs t, u] | 10 | n/a | n/a |
| *Basal face* | Si-O-Si, Al-O-Si [ref e] | < 4 [refs l, n] | | | | |
| *Edges* | Si-OH, SiAl-O$^{1/2-}$, Al$_2$-OH, Al-OH$^{1/2-}$, Al-OH$_2^{1/2+}$ [ref e] | 7 – 8 [refs l, n] | | | | |
| **Gibbsite** | Al-OH$^{1/2-}$, Al-OH$_2^{1/2+}$, Al$_2$-OH [ref f] | 7.5 – 11.3 [ref o] | $10^{-14} - 10^{-13}$ at pH ~7 [ref v] | 10 | n/a | n/a |
| **Quartz** | Si-O$^-$, Si-OH, Si-(OH)$_2$, Si-(OH)O$^-$ [ref g] | 2 [ref h] | $10^{-13} - 10^{-12}$ at pH ~7 [refs w, x] | 1 - 9 | Milled: 247-251 | 0.7 – 0.92 |
| | | | | 5 | Long-term aged: ~239 K | ~0.1 |

a: Teng et al., (2001); b: Schoonheydt and Johnston, (2006); c: Brady et al. (1996); d: Liu et al. (2013); e: Yan et al. (2011); f: Hiemstra et al. (1999); g: Liu et al. (2014) ; h: Vidyadhar and Hanumantha Rao (2007); i: Karagüzel et al. (2005); k: Gupta and Miller (2010); l: Kumar et al. (2017); m: Liu et al. (2014); n: Zhao et al. (2008); o: Kosmulski (2009); p : Crundwell (2015); q : Gudbrandsson et al. (2014); r : Carroll and Walther (1990); s: Huertas et al. (1999); t: Brady and Walther (1989); u: Lammers et al. (2017); v: Dietzel and Böhme (2005); w: Brady and Walther (1990); x: Berger et al. (1994); n/a: no $T_{het}$ and $F_{het}$ can be reported due to the absence of a discernible heterogeneous freezing signal in the emulsion freezing experiments



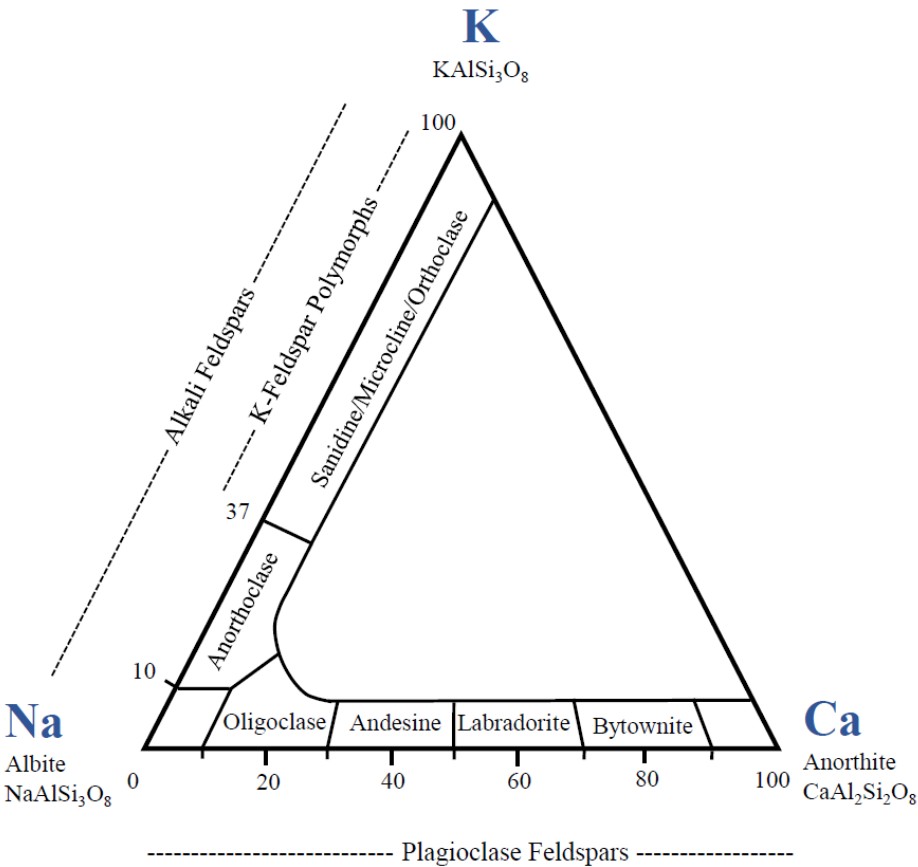

**Figure 1.** Feldspar classification diagram. Numbers are contribution of the end-member minerals in wt% (adapted from Greenwood and Earnshaw (1998)).

50



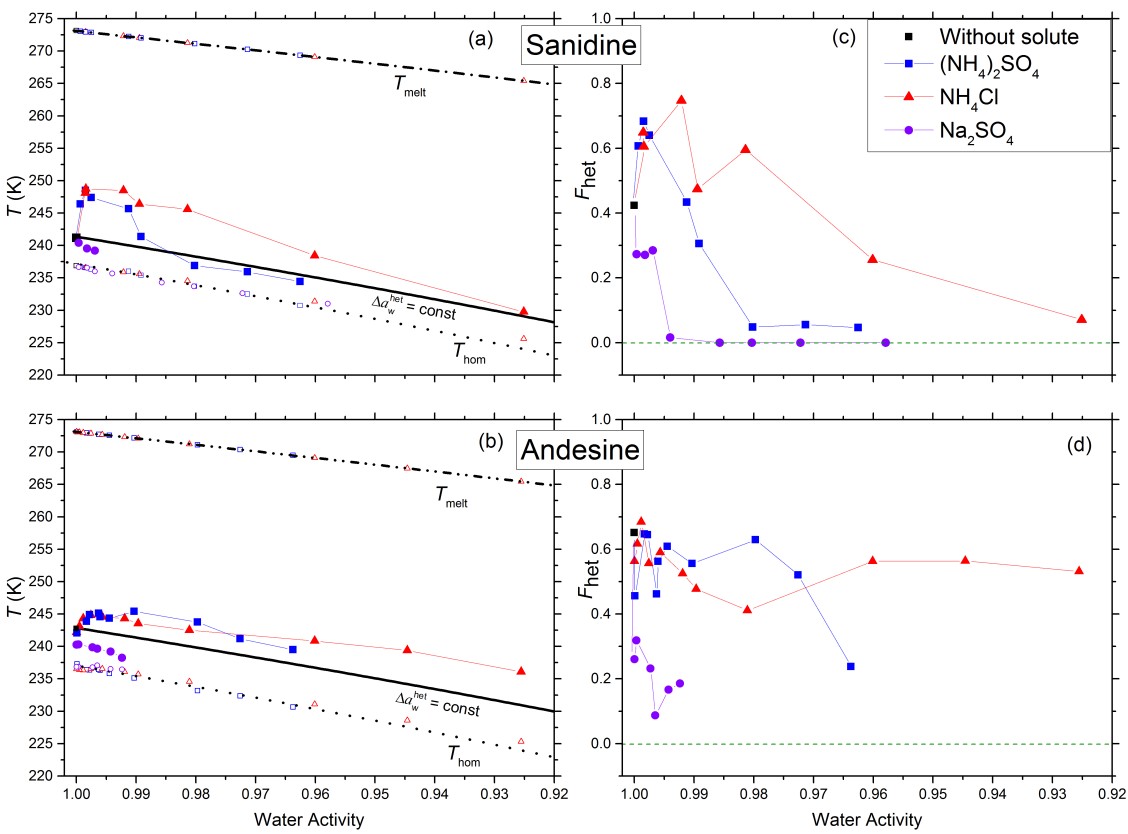

**Figure 2.** Measured freezing temperatures (left) and heterogeneously frozen fractions (right) of emulsion freezing experiments
with 2 wt % feldspar, namely sanidine (panels a and c) and andesine (panels b and d), in various solutions (color-coded). **Left:**
Heterogeneous freezing onset temperatures, $T_{het}$ (filled solid symbols connected by thin lines to guide the eye), homogeneous
freezing onset temperatures, $T_{hom}$ (open symbols at $T < 237$ K), and ice melting temperatures, $T_{melt}$ (open symbols at $T > 265$ K)
as functions of water activity of solutions, $a_w$, for various solutes (symbols and colors see insert). Dash-dotted black line: ice
melting point curve. Dotted black line: homogeneous ice freezing curve for supercooled aqueous solutions obtained by horizon-
tally shifting the ice melting curve by a constant offset $\Delta a_w^{hom}(T) = 0.294$. Solid black line: horizontally shifted from the ice
melting curve by $\Delta a_w^{het}(T) = 0.264$ and $0.254$ for sanidine (panel a) and andesine (panel b), respectively, with offsets derived
from the heterogeneous freezing temperature of the suspension of the mineral in pure water (filled black square at $a_w = 1$). Sym-
bols are the mean of at least two separate emulsion freezing experiments. Difference between the two measurements plus instru-
mental uncertainty in $T_{het}$ and $a_w$ are smaller than the symbol size (see Sect 2.2). **Right:** Heterogeneously frozen fraction, $F_{het}$, as
function of $a_w$, for sanidine (panel c) and andesine (panel d). Measurement difference and uncertainty in $F_{het}$ do not exceed $\pm 0.1$.



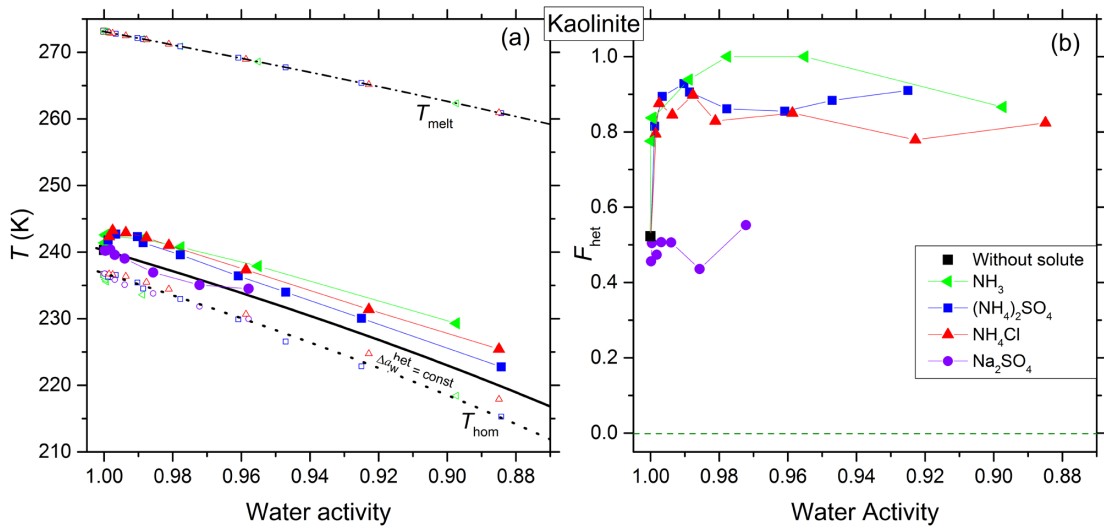

**Figure 3.** Same as Fig. 2, but for kaolinite (5 wt %). (a) Homogeneous ice freezing curve (dotted black line) characterized by a
constant offset $\Delta a_w^{hom}(T) = 0.294$. Ice melting curve (dash-dotted black line) and the heterogeneous freezing curve (solid black
line) horizontally shifted from the ice melting curve by $\Delta a_w^{het}(T) = 0.272$ derived from the heterogeneous freezing temperature of
the suspension of the mineral in pure water (filled black square at $a_w = 1$). Measured differences and instrumental uncertainties in
$T_{het}$ and $a_w$ are smaller than the symbol size (see Sect. 2.2). (b) $F_{het}$ (heterogeneously frozen fraction) as function of water activity
of the solutions. Measurement difference and uncertainty in $F_{het}$ do not exceed ± 0.1.



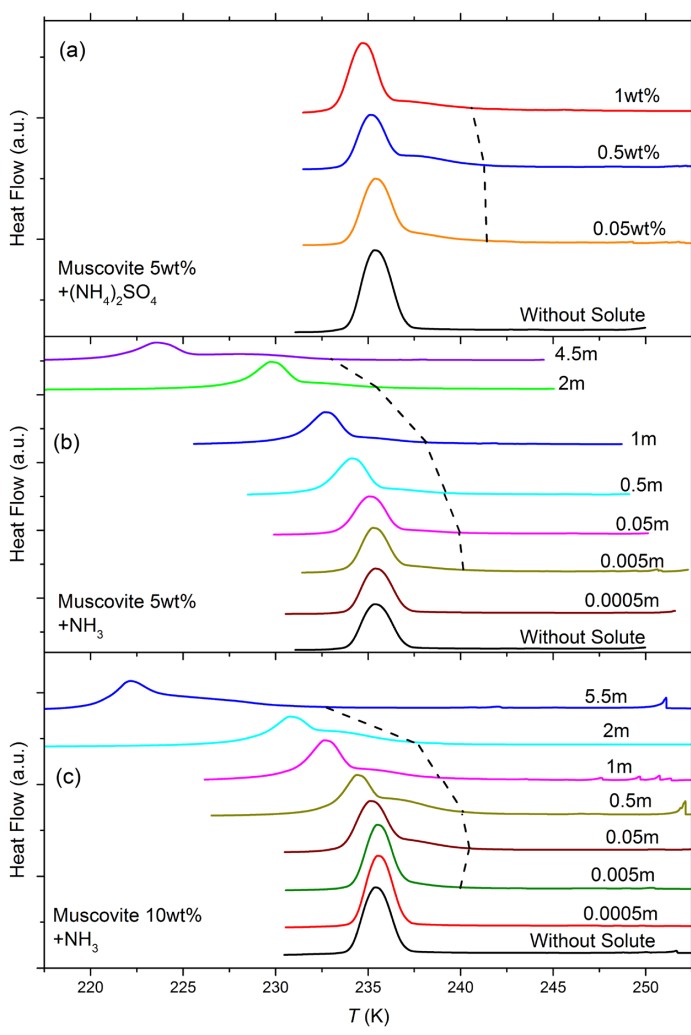

**Figure 4.** DSC thermograms of 5 wt % (panels a and b) and 10 wt% (panel c) muscovite particles suspended in ammonia (0 - 5.5 molal; $a_w = 0.999 - 0.912$) and ammonium sulfate (0 − 1 wt %; $a_w = 0.996 - 0.988$) droplets of different solution concentrations (given as inserts). All curves are normalized such that the areas under the heterogeneous and homogeneous freezing curves sum up to the same value. The dashed black line connects the heterogeneous freezing onset temperatures ($T_{het}$) of the emulsions. With increasing ammonia concentration heterogeneous IN efficiency starts to develop, as can be noticed from the appearance of $T_{het}$ which was absent in pure water. The intensity of the heterogeneous freezing signal, hence $F_{het}$ becomes more prominent in 10 wt % suspensions (see Table 1).





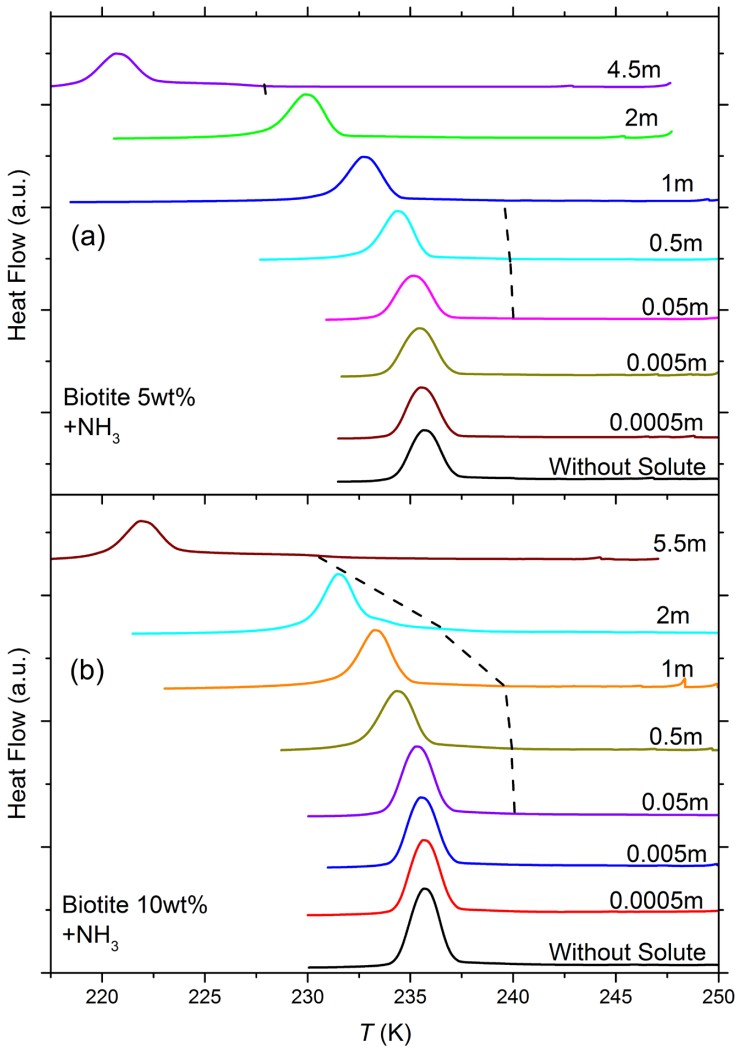

**Figure 5.** DSC thermograms of 5 wt % (panel a) and 10 wt % (panel b) biotite particles suspended in ammonia solution droplets (0 - 5.5 molal; $a_w$ = 0.999 – 0.912). All curves are normalized such that the areas under the heterogeneous and homogeneous freezing curves sum up to the same value. In contrast to muscovite, biotite becomes IN active only at very high concentrations of ammonia. In addition, the intensity of the heterogeneous freezing signal (hence $F_{het}$) is low, but becomes more prominent in 10 wt % suspensions (see Table 2).



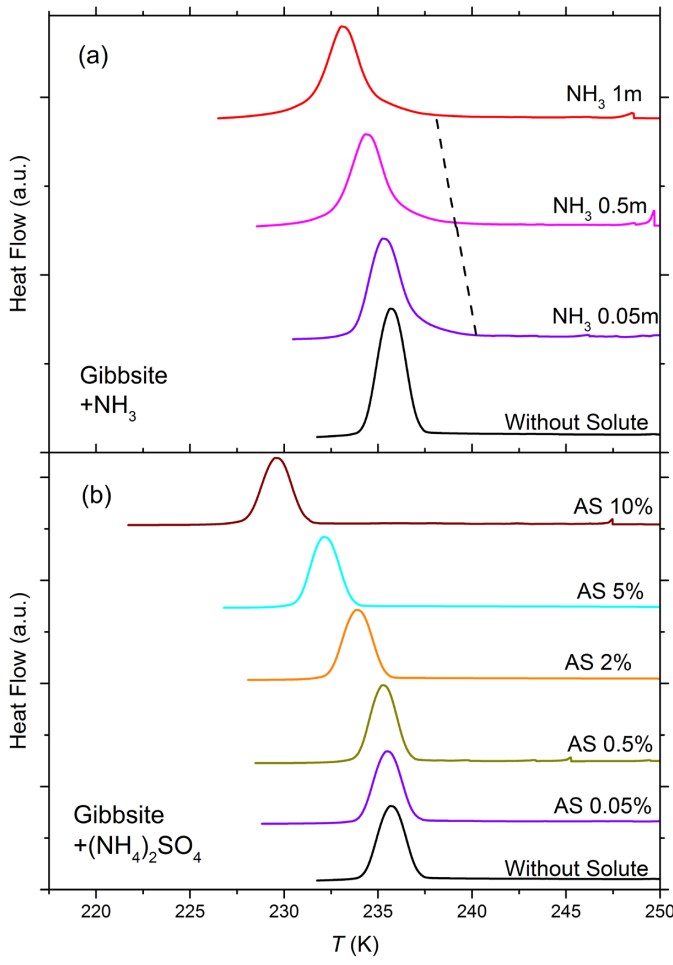

**Figure 6.** DSC thermograms of 10 wt % gibbsite suspended in ammonia solution droplets (panel a) of concentrations from 0 to 1 molal ($a_w = 1 - 0.981$) and $(NH_4)_2SO_4$ solution droplets (panel b) from 0 to 10 wt % ($a_w = 1 - 0.961$). All curves are normalized such that the areas under the heterogeneous and homogeneous freezing curves sum up to the same value. Gibbsite shows no discernible heterogeneous freezing signal in pure water and starts to show a weak heterogeneous freezing signal when suspended in 100 ammonia solution (dashed black line connects the onset of heterogeneous freezing signals), while no heterogeneous IN is observed for gibbsite suspended in $(NH_4)_2SO_4$.





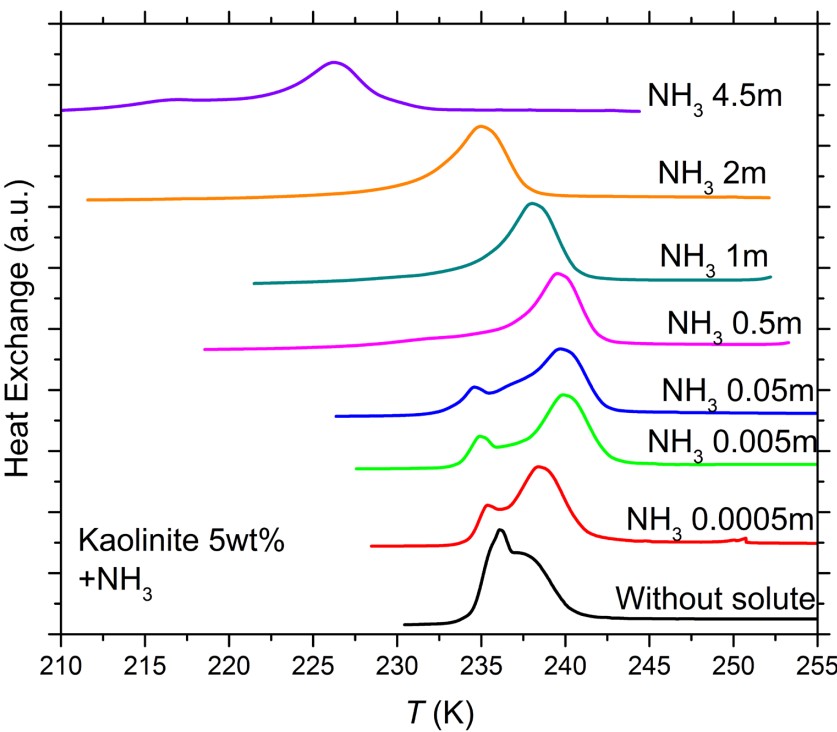

**Figure 7.** DSC thermograms of 5 wt % kaolinite particles suspended in ammonia solution droplets (0 - 4.5 molal; $a_w$ = 1 – 0.922). All curves are normalized such that the areas under the heterogeneous and homogeneous freezing curves sum up to the same value.



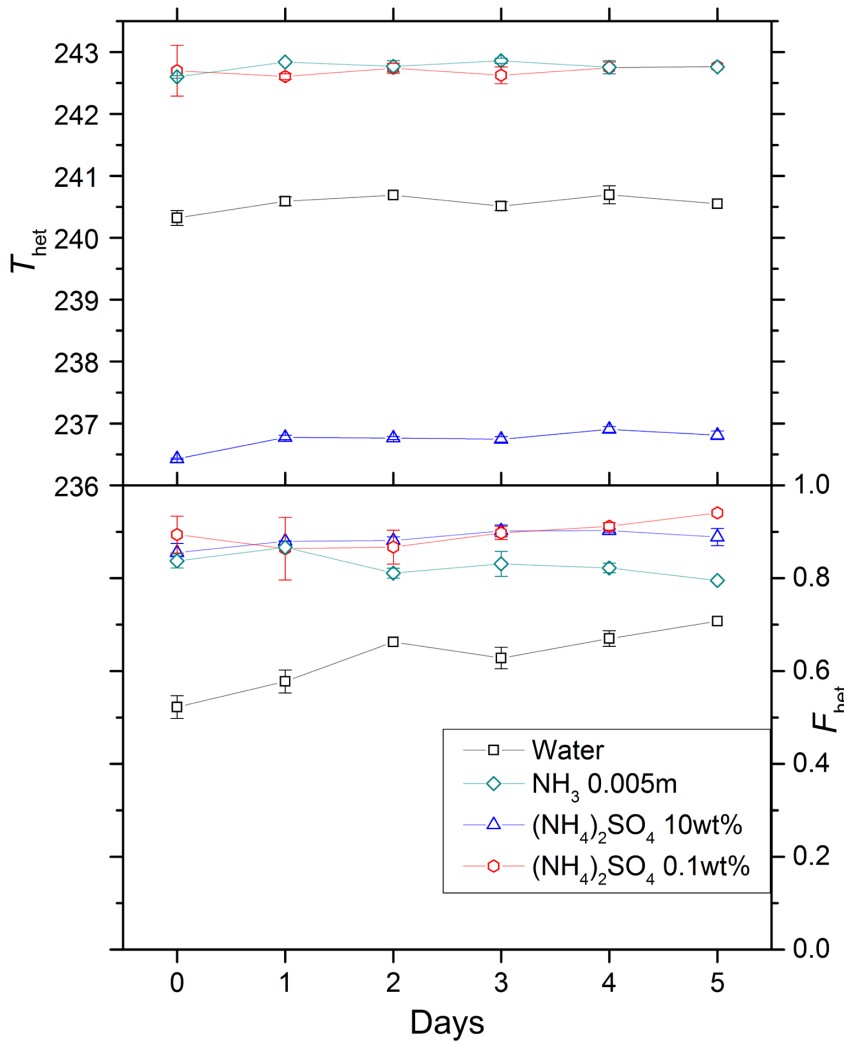

**Figure 8.** Development of $T_{het}$ (upper panel) and $F_{het}$ (lower panel) for 5 wt % kaolinite suspended in water, 10 wt % $(NH_4)_2SO_4$, 0.1 wt % $(NH_4)_2SO_4$, 0.005 molal ammonia solutions over a period of 5 days. Data points depict mean (with error bars representing min-to-max) for $T_{het}$ and $F_{het}$ measured for two aging experiments performed with two independently aged suspensions.



**Minerals:** Feldspars (microcline, sanidine, andesine), clay mineral (kaolinite), micas (muscovite, biotite), aluminum hydroxide (gibbsite), quartz.

**Mineral features and processes:** (i) Isomorphic substitution leads to permanent surface charge, neutralized by cations; (ii) surface OH groups form H-bond with water molecules; (iii) protonation/deprotonation of OH groups imparts pH-dependent surface charge; (iv) H-bonding of $NH_3$ with surface OH groups provides a better template for ice formation

**1. Enhanced $T_{het}$ of feldspars in dilute $NH_3/NH_4^+$ suspensions.**
Likely reason: bonding of $NH_3$ with surface OH groups provides a better template for ice formation.

**2. Alkali salts hamper IN activity of feldspars.**
Likely reason: $H^+/H_3O^+$ exchange with surface cations, essential for IN ability in water, is suppressed by excess alkali ions.

**3. Alkali salts do not affect IN activity of kaolinite.**
Likely reason: Ion exchange is absent in kaolinite, hence no effect. Yet, bonding of $NH_3$ with surface still provides a better template for ice formation.

**4. Activation of micas + gibbsite in dilute $NH_3/NH_4^+$ suspensions.**
Likely reason: Showing no IN activity in pure water, bonding of $NH_3$ with surface makes micas and gibbsite IN active.

**5. $NH_3$ hampers IN on quartz.**
Likely reason: Enhanced dissolution under alkaline conditions.

**6. Milling makes quartz IN active.**
Likely reason: Milling-induced defects serve as active sites.

**7. Quartz becomes inactive by long-term aging in water.**
Likely reason: Defects on quartz surface vanish after months by self-healing.

**Figure 9.** Annotated summary: heterogeneous IN onset temperatures, $T_{het}$, of various mineral dusts (2 − 10 wt%) as function of water activity in $NH_3$, $(NH_4)_2SO_4$ and $Na_2SO_4$ solutions (color-coded) for $a_w = 1.0 − 0.92$. (a) Results for microcline from Kumar et al. (2018a). (b)-(g) Feldspars, kaolinite, micas and gibbsite investigated in the present study. (h) Results for quartz from Kumar et al. (2018b). Dash-dotted black lines: ice melting point curves. Dotted black lines: homogeneous ice freezing curves for supercooled aqueous solutions. Solid black lines: expected heterogeneous freezing curves, if the presence of solutes did not change the mineral surface properties. Lines result by horizontally shifting from the ice melting curve by a constant $\Delta a_w^{het}$ emanating from the heterogeneous freezing temperatures of the suspensions of the minerals in pure water (at $a_w = 1$). Specifically, $\Delta a_w^{microcline} = 0.296$, $\Delta a_w^{sanidine} = 0.264$, $\Delta a_w^{andesine} = 0.254$, $\Delta a_w^{kaolinite} = 0.272$, $\Delta a_w^{quartz} = 0.221$.