# Peer review of "Ice nucleation activity of silicates and aluminosilicates in pure water and aqueous solutions. Part 3 – Aluminosilicates"

_Atmospheric Chemistry and Physics, 2018_

## Referee Comment (RC1) · Anonymous Referee #1 · 20 Dec 2018

**Review of "Ice nucleation activity of silicates and aluminosilicates in pure water and aqueous solutions. Part 3 – Aluminosilicates" by Kumar et al.**

**General Comment:**

This is the third manuscript of a series of studies that aimed to understand the ice nucleating abilities of mineral dust particles with an especial focus on feldspars. Part 3 reports the ice nucleating abilities of different aluminosilicate particles in the immersion freezing mode in pure water and aqueous solutions. The authors investigated the effect of surface ion exchange, $NH_3$ or $NH_4^+$ adsorption, and surface degradation on the ice nucleating abilities of kaolinite, sanidine, andesine, muscovite, biotite and gibbsite. The ice nucleating abilities were enhanced in some cases and reduced in others. The increase/decrease of the ice nucleating abilities was found to depend on the type of mineral and also on the exposure time of the surfaces to water and solutes.

This is a very interesting topic of high relevance for the atmospheric chemistry and physics communities. The experiments from the current study were carefully designed and performed. The current results brings our understanding one step forward and it helps the ice nucleation community to better understand why mineral dust particles are good ice nucleating particles. Although the reviewer did not find any major point in terms of the scientific content and the drawn conclusions, the structure, length, and readability of the manuscript needs to be significantly improved before it can be accepted for its publication in ACP.

**"Major" comments:**

1. The manuscript is unnecessary long and different information is repeated several times along the text. This makes the manuscript difficult to read and confusing in some parts.
2. The structure of the manuscript is not the best with too many subsections and mixing information from sections 3 and 4.
3. The usage of qualitative terms (e.g., strong decrease, remarkable enhancement, strong enhancement) is pronounced and not appropriate.

**Minor comments:**

1. Abstract: I suggest to reduce its length and to focus on the main results only. Leave out unnecessary or theoretical information for the Introduction section.

2. Introduction: It is missing why the authors focused on immersion freezing and what is the importance of mixed-phase clouds. Also, why are mineral dust particles important on a global or regional scale, and why this specific aerosol type is important in comparison to biological or organic particles?

3. There are too many statements/paragraphs without a proper citation(s).

**Specific comments:**

Line 38: Add a reference after "properties".

Line 41: Add a reference after "crystalline ice".

Line 46: Add a reference after "deposition nucleation".

Lines 52-53: Either clarify that this studies refer to immersion freezing only or add studies for other ice nucleation modes using feldspar.

Line 56: Add the Kanji et al. ACPD (2018) study.

Line 58: Add a reference after "humidity".

Line 62: Add a reference after "function of $a_w$".

Line 64: Add a reference after "solute".

Line 73: Add "previous" after "our" and before "freezing".

Line 84: Why immersion freezing? Why is this heterogeneous freezing mode important?

Lines 90-101: This paragraph deserves more than a single reference.

Line 103: Add a reference after "time".

Line 107: Add a reference after "crust".

Line 111: Add a reference after "edges".

Line 129: Add a reference after "dusts".

Line 167: Add a reference after "1000 particles".

Lines 192: Given that the authors separated the Results from the Discussion of the Results in different sections, there are several parts which are repeated. It would be ideal to reduce this where possible. I suggest combining the Results and Discussion sections to avoid redundancy along the manuscript and to reduce its length.

Line 201: Add a reference after "ice".

Line 219, 228: What is strong? I suggest to report this in a more statistical fashion.

Lines 223-224, 241: "Remarkable enhancement". Please be more specific.

Line 241: "Strong enhancement". Please be more specific.

Line 293: "because of the high bonding energy involved". On the other of?

Line 330: "a thick amorphous surface". How thick?

Lines 383-384: Given that there are many studies on this topic, I suggest to include review papers only where most of previous studies are included (e.g., Hoose and Mohler (2012), Murray et al. (2012), and Kanji et al. (2017)).

Line 385: Is this sample only used by ETH research groups?

Line 385: For consistency also mentioned relevant studies using the KGa-1b sample.

Lines 390-391: Why this should not be the case if both studies used the same sample and the same instrument?

Lines 391-395: I do not get the point or contribution of this paragraph.

Lines 410-444: This is a nice review of recent studies and discoveries on this topic, but this is unnecessary long and I believe that it does not contribute too much to the manuscript. I suggest to summarize this and to focus on the papers that are strictly necessary to explain the present results.

Line 368: This belongs to section 3 instead to section 4.

Line 486: Line 201: Add a reference after "activity".

Lines 510-549: Unnecessary long. Please be more concise and specific.

Lines 550-634: This is a super long summary and it repeats many things already mentioned in the Abstract and Section 4. If the authors would like to keep this section focus on the main results/discoveries/conclusions only.

Figures 2 and 3: Open symbols are too small.

---

## Referee Comment (RC2) · Anonymous Referee #2 · 25 Jan 2019

General comments

Kumar et al report on the ice nucleation activity of a range of aluminosilicate minerals and on the impact of various solute molecules on this activity. It is concluded that exchange of native cations with protons probably plays a critical role in the ice nucleating ability of aluminosilicates and that this explains why alkali salts inhibit ice nucleation by some aluminosiliciates. Similarly, it is concluded that that NH3 and NH4+ adsorb to the surface of feldspars rather than exchanging with cations and from there interact with water in a way that promotes ice nucleation, and that dissolution of feldspar surfaces and formation of an amorphous surface layer inhibits ice nucleation by feldspars.

[Figure]

The paper also summarises the more numerous findings of the entire 3 part series of papers, concluding that relevant factors for the IN activity of silicates and aluminosilicates more generally include adsorption and ion exchange with the mineral surfaces and changes to surface structures induced by dissolution and growth of mineral surfaces exposed to water. The authors are sensibly tentative about their conclusions and suggest that molecular dynamics simulations and surface science techniques may allow them to be tested more thoroughly.

The paper contains many results and reasonable interpretations. I am sure it will be of substantial interest to the audience of ACP, and possibly to those interested in heterogeneous nucleation more generally.

While I support publication, I note that the paper is long and complicated. While mostly well written, it is very hard to read and digest. It would certainly be beneficial if its bulk were reduced and the information it contains summarised more concisely. While I appreciate the difficulty in being more concise when dealing with so many different results I would strongly recommend that some effort be made to address this issue before publication. I also have some more specific scientific comments that the authors may wish to consider.

Specific comments

I have some thoughts on the interpretation of results concerning the feldspars. Feldspar mineralogy is complex compared to that of the other minerals investigated, is clearly of substantial relevance to the topic in question but is not, to my mind, dealt with thoroughly in this manuscript.

Characterisation of the feldspar samples is slightly lacking. What is the chemical composition of the three samples used? That the plagioclase sample is named 'andesine' implies a composition but no precise composition is given. The sanidine and microcline samples used could have any composition between 37% K and 100% K from what has been presented. Kauffman et al. (2016) does not contain this information either, as far

as I can tell and also does not have an 'andesine' sample. I do not think it is reasonable to assume that the ice nucleating ability of the samples used are representative of their crystallographic structures. Harrison et al. (2016) and Whale et al. (2017) observe substantial variability in the ice nucleation effectiveness between crystallographically similar feldspars. Notably, Harrison et al. tested a sanidine sample that nucleated ice with similar effectiveness to most microclines and Whale et al. confirmed that it was sanidine using Raman spectroscopy. The argument is put forward here that the sorts of features that Whale et al. hypothesise cause ice nucleation in larger water droplets are not likely to be present in the smaller droplets used for this study. I am not sure about this. It is quite clear that different alkali feldspars have different structures across multiple scales, including the nanometer scale (Parsons et al., 2015). This is relevant because a) nanoscale features could easily be present on the 'submicrometer' parti-cles used in this paper and b) nanoscale features are most probably on a scale similar to the critical ice nucleus.

It is a little hard to pick out but it appears that the authors think that the plagioclase and sanidine feldspar they have tested are less active than microcline because the sites present on these feldspars are dissolved away effectively immediately on contact with water. Feldspar dissolution is clearly a complex topic. I do not feel I am qualified to comment on the authors' interpretation of this literature but I would note that it does not appear to be a well settled and understood subject from what I have read. It seems likely that feldspar-to-feldspar variability in dissolution rate is associated with more than just crystal structure and stoichiometry.

Still it is helpful that opposing hypotheses for the differing ice nucleation abilities of feldspars now exist, as it is probably straightforward to test which is more consistent with experiments. It would, for instance, be interesting to see if the LD2 sanidine of Harrison et al. and Whale et al. nucleates ice better than a micro-texturally pristine Eifel sanidine (which is what I expect has been tested in this study) in the smaller emulsion droplets.

The citation for the dissolution rates of feldspars is wrong. I think the authors have read the numbers used from graphs in 'The mechanism of dissolution of the feldspars: Part I. Dissolution at conditions far from equilibrium' but have cited part IV.

References

Harrison, A. D., Whale, T. F., Carpenter, M. A., Holden, M. A., Neve, L., O'Sullivan, D., Vergara Temprado, J., and Murray, B. J.: Not all feldspars are equal: a survey of ice nucleating properties across the feldspar group of minerals, Atmos. Chem. Phys., 16, 10927-10940, 10.5194/acp-16-10927-2016, 2016. Kaufmann, L., Marcolli, C., Hofer, J., Pinti, V., Hoyle, C. R., and Peter, T.: Ice nucleation efficiency of natural dust samples in the immersion mode, Atmos. Chem. Phys., 16, 11177-11206, 10.5194/acp-16-11177-2016, 2016. Parsons, I., Fitz Gerald, J. D., and Lee, M. R.: Routine characterization and interpretation of complex alkali feldspar intergrowths, Am. Mineral., 100, 1277-1303, 10.2138/am-2015-5094, 2015. Whale, T. F., Holden, M. A., Kulak, A. N., Kim, Y.-Y., Meldrum, F. C., Christenson, H. K., and Murray, B. J.: The role of phase separation and related topography in the exceptional ice-nucleating ability of alkali feldspars, Phys. Chem. Chem. Phys., 10.1039/C7CP04898J, 2017.

---

## Author Comment (AC1) · 25 Mar 2019

We thank Reviewer 1 for his/her constructive comments. We reproduce reviewer's comments in *blue* and our responses in black.

*General Comment:*

*This is the third manuscript of a series of studies that aimed to understand the ice nucleating abilities of mineral dust particles with an especial focus on feldspars. Part 3 reports the ice nucleating abilities of different aluminosilicate particles in the immersion freezing mode in pure water and aqueous solutions. The authors investigated the effect of surface ion exchange, $NH_3$ or $NH_4^+$ adsorption, and surface degradation on the ice nucleating abilities of kaolinite, sanidine, andesine, muscovite, biotite and gibbsite. The ice nucleating abilities were enhanced in some cases and reduced in others. The increase/decrease of the ice nucleating abilities was found to depend on the type of mineral and also on the exposure time of the surfaces to water and solutes.*

*This is a very interesting topic of high relevance for the atmospheric chemistry and physics communities. The experiments from the current study were carefully designed and performed. The current results brings our understanding one step forward and it helps the ice nucleation community to better understand why mineral dust particles are good ice nucleating particles. Although the reviewer did not find any major point in terms of the scientific content and the drawn conclusions, the structure, length, and readability of the manuscript needs to be significantly improved before it can be accepted for its publication in ACP.*

*Major comments:*

*The manuscript is unnecessary long and different information is repeated several times along the text. This makes the manuscript difficult to read and confusing in some parts.*

We have made appropriate changes throughout the manuscript in order to remove the repeated information.

*The structure of the manuscript is not the best with too many subsections and mixing information from sections 3 and 4.*

We have restructured the manuscript by combining the results and discussion sections in a "*Results and discussion*" section, which is now ordered according to minerals.

*The usage of qualitative terms (e.g., strong decrease, remarkable enhancement, strong enhancement) is pronounced and not appropriate.*

We have made appropriate changes throughout the manuscript in order to reduce the usage of such terms.

*Minor comments:*

*Abstract: I suggest to reduce its length and to focus on the main results only. Leave out unnecessary or theoretical information for the Introduction section.*

The 1$^{st}$ part of the abstract summarizes the results of Part 3 and the 2$^{nd}$ part gives the synopsis of the whole paper series. We would like to keep both parts since it provides a quick summary of the series for the readers. We have shortened the abstract slightly by removing some less important text.

*Introduction: It is missing why the authors focused on immersion freezing and what is the importance of mixed-phase clouds. Also, why are mineral dust particles important on a global or regional scale, and why this specific aerosol type is important in comparison to biological or organic particles?*

We have added the following statements in the *Introduction* to support the points highlighted by the reviewer.

"Moreover, mixed-phase clouds are responsible for precipitation formation via the Wegener-Bergeron-Findeisen process (Rogers and Yau, 1989; Korolev and Field, 2008)." (Lines 38-39)

"The most important ice nucleation mechanisms in mixed-phase clouds are viewed to be immersion and condensation freezing (Hoose et al., 2008; Ansmann et al., 2009; Twohy et al., 2010; de Boer et al., 2011)." (Lines 50-51)

"The impact of mineral dusts on cloud properties has been shown in several observational and modelling studies (Lohmann and Diehl, 2006; Hoose et al., 2010; Seifert et al., 2010)." (Lines 54-55)

*There are too many statements/paragraphs without a proper citation(s).*
*Specific comments:*
*Line 38: Add a reference after "properties".*
Sun and Shine (1994); Sassen and Benson (2001) have been added as references. (Line 37)

*Line 41: Add a reference after "crystalline ice".*
Vali et al. (2015) has been added as reference. (Line 42)

*Line 46: Add a reference after "deposition nucleation".*
Vali et al. (2015) has been added as reference. (Line 47)

*Lines 52-53: Either clarify that this studies refer to immersion freezing only or add studies for other ice nucleation modes using feldspar.*
Suggested lines have been modified to: "Feldspars have been reported to be the most IN active minerals although the individual members of the feldspar group exhibit very different IN efficiencies in immersion freezing mode." (Lines 55-57)

*Line 56: Add the Kanji et al. ACPD (2018) study.*
Kanji et al. (2018) has been added. (Line 60)

*Line 58: Add a reference after "humidity".*
Augustin-Bauditz et al. (2014) has been added. (Line 62)

*Line 62: Add a reference after "function of aw".*
The references to this statement are mentioned 2 lines below "…INPs can indeed be approximated by such a water-activity-based description (Archuleta et al., 2005; Zobrist et al., 2006; Zobrist et al., 2008; Koop and Zobrist, 2009; Knopf et al., 2011; Knopf and Forrester, 2011; Knopf and Alpert, 2013; Rigg et al., 2013)." (Lines 68-70)

*Line 64: Add a reference after "solute".*
Zobrist et al. (2008) has been added. (Line 68)

*Line 73: Add "previous" after "our" and before "freezing".*
Added (Line 78)

*Line 84: Why immersion freezing? Why is this heterogeneous freezing mode important?*

We have added the importance of immersion freezing in the *Introduction* as previously suggested by the reviewer. (Line 50)

*Lines 90-101: This paragraph deserves more than a single reference.*

Two more references have been added: Brown and Parsons (1989) and Hofmeister and Rossman (1983) (Lines 163, 166)

*Line 103: Add a reference after "time".*

This paragraph has been removed.

*Line 107: Add a reference after "crust".*

Murray (1991) added. (Line 311)

*Line 111: Add a reference after "edges".*

Bibi et al. (2016) added. (Line 316)

*Line 129: Add a reference after "dusts".*

Boose et al. (2016) and Kaufmann et al. (2016) added.

*Line 167: Add a reference after "1000 particles".*

Kumar et al. (2018a) added. (Line 132)

*Lines 192: Given that the authors separated the Results from the Discussion of the Results in different sections, there are several parts which are repeated. It would be ideal to reduce this where possible. I suggest combining the Results and Discussion sections to avoid redundancy along the manuscript and to reduce its length.*

We have combined the results and discussion sections in a "*Results and discussion*" section, which is now ordered according to minerals. As suggested by the reviewer, we have also removed less important text to shorten the *Results* and *Discussion* section.

*Line 201: Add a reference after "ice".*

Negi and Anand (1985) added. (Line 180)

*Line 219, 228: What is strong? I suggest to report this in a more statistical fashion.*

We have replaced the whole sentence with: "In contrast to $NH_3/NH_4^+$-solutions, freezing experiments in the presence of $Na_2SO_4$ as a non-$NH_4^+$ solute show a decrease in $T_{het}$ below $T_{het}^{\Delta a_w}(a_w)$ by ~1.7 K and ~2.4 K for sanidine and andesine, respectively, at $a_w \approx 0.99$. No discernible heterogeneous freezing signal was observed for higher $Na_2SO_4$ concentrations ($a_w < 0.99$)." (Lines 198-200)

*Lines 223-224, 241: "Remarkable enhancement". Please be more specific.*

We have replaced the whole sentence with: "In dilute $NH_4^+$-containing solutions, sanidine shows an enhancement in $F_{het}$ up to ~0.75 at $a_w \approx 0.99$ compared to the suspension in pure water ($F_{het} = 0.42$ at $a_w = 1$)." (Lines 202-203)

*Line 241: "Strong enhancement". Please be more specific.*
We revised it to: "Interestingly, kaolinite shows a strong enhancement of $F_{het}$ to $0.75 - 1.00$ compared to the pure water case ($F_{het} = 0.52$) in the presence of $NH_3$ and $NH_4^+$-solutes over the complete investigated concentration range ($a_w = 1 - 0.88$)." (Lines 333-335)

*Line 293: "because of the high bonding energy involved". On the other of?*
Nash and Marshall (1957) reported that a part of the ammonium ions is firmly fixed to the surface because they could not be exchanged by other cations. However, they did not give any number for the bonding energy. We re-phrase the sentence to follow better Nash and Marshall (1957): "Ammonium ions not only have a strong preference for cation exchange with K-feldspars and (Na-Ca)-feldspars but part of them remain fixed to the surface in non-exchangeable form." (Lines 244-246)

*Line 330: "a thick amorphous surface". How thick?*
We have re-phrased the line to: "….we hypothesize that amorphous surface layers exceeding few nanometers in depth hamper the IN activity of feldspars." (Lines 280-281)

*Lines 383-384: Given that there are many studies on this topic, I suggest to include review papers only where most of previous studies are included (e.g., Hoose and Mohler (2012), Murray et al. (2012), and Kanji et al. (2017)).*
Citations have been modified to Hoose and Mohler (2012), Murray et al. (2012) and Kanji et al. (2017). (Line 344)

*Line 385: Is this sample only used by ETH research groups?*
KGa-1b has been used by several research groups. It is a relatively pure kaolinite provided by Clay Mineral Society and has a low density of crystal lattice defects. We make this clear in the revised manuscript by stating: "… while others used the KGa-1b kaolinite from the Clay Mineral Society with high mineralogical purity (96 %, minor impurities of anatase, crandallite, mica, and illite) (Murray et al., 2011; Pinti et al., 2012; Wex et al., 2014; Kaufmann et al., 2016)". (Lines 345-347)

*Line 385: For consistency also mentioned relevant studies using the KGa-1b sample.*
Murray et al. (2011); Pinti et al. (2012); Wex et al. (2014); Kaufmann et al. (2016) added. (Lines 347)

*Lines 390-391: Why this should not be the case if both studies used the same sample and the same instrument?*
This is indeed what is expected, nevertheless, the IN efficiency could change due to storage over longer time periods depending on storage conditions.

*Lines 391-395: I do not get the point or contribution of this paragraph.*
We have deleted the suggested part to shorten the manuscript.

*Lines 410-444: This is a nice review of recent studies and discoveries on this topic, but this is unnecessary long and I believe that it does not contribute too much to the manuscript. I suggest to summarize this and to focus on the papers that are strictly necessary to explain the present results.*

Lines 410 – 433 (lines 360-394 in the revised manuscript) characterize the surface functional groups present on the different kaolinite surfaces. We think that this is relevant information summarized here from studies comprising a broad range of sources, which should become accessible to our community.

Lines 434 – 448 (lines 384-398 in the revised manuscript) summarize molecular dynamics studies carried out with different kaolinite surfaces, which might be deleted but we think that readers that are not interested could also just skip this part.

*Line 368: This belongs to section 3 instead to section 4.*

Sections 3 and 4 from the previous manuscript are now merged together into one section "*Results and discussion*".

*Line 486: Line 201: Add a reference after "activity".*

Kumar et al. (2018b) added. (Line 434)

*Lines 510-549: Unnecessary long. Please be more concise and specific.*

We prefer to keep this part since it summarizes surface properties of micas collected from various studies. We think that all available information about mineral surfaces need to be considered as long as it is not clear which are the relevant ones for ice nucleation.

*Lines 550-634: This is a super long summary and it repeats many things already mentioned in the Abstract and Section 4. If the authors would like to keep this section focus on the main results/discoveries/conclusions only.*

This section is a synopsis of the main findings of Parts 1 – 3 of this series and brings them in a broader context. It shows e.g. that milling is highly relevant for silica but not for feldspars and that the enhancing effect of $NH_3/NH_4^+$ pertains to feldspars and clay minerals but not to silica. We think that it is highly relevant to show that results obtained for one mineral should not be generalized to all minerals. We critically went through the whole section and shortened some parts, while we would like to keep most of it.

*Figures 2 and 3: Open symbols are too small.*

The size of open symbols in Figures 2 and 3 have been increased in the revised manuscript.

**References**

[revised manuscript text omitted]

---

## Author Comment (AC2) · 25 Mar 2019

We thank Reviewer 2 for his/her constructive comments. We reproduce reviewer's comments in *blue* and our responses in black.

*General comments*

*Kumar et al report on the ice nucleation activity of a range of aluminosilicate minerals and on the impact of various solute molecules on this activity. It is concluded that exchange of native cations with protons probably plays a critical role in the ice nucleating ability of aluminosilicates and that this explains why alkali salts inhibit ice nucleation by some aluminosilicates. Similarly, it is concluded that that NH3 and NH4+ adsorb to the surface of feldspars rather than exchanging with cations and from there interact with water in a way that promotes ice nucleation, and that dissolution of feldspar surfaces and formation of an amorphous surface layer inhibits ice nucleation by feldspars. The paper also summarizes the more numerous findings of the entire 3 part series of papers, concluding that relevant factors for the IN activity of silicates and aluminosilicates more generally include adsorption and ion exchange with the mineral surfaces and changes to surface structures induced by dissolution and growth of mineral surfaces exposed to water. The authors are sensibly tentative about their conclusions and suggest that molecular dynamics simulations and surface science techniques may allow them to be tested more thoroughly. The paper contains many results and reasonable interpretations. I am sure it will be of substantial interest to the audience of ACP, and possibly to those interested in heterogeneous nucleation more generally.*

*While I support publication, I note that the paper is long and complicated. While mostly well written, it is very hard to read and digest. It would certainly be beneficial if its bulk were reduced and the information it contains summarized more concisely. While I appreciate the difficulty in being more concise when dealing with so many different results I would strongly recommend that some effort be made to address this issue before publication. I also have some more specific scientific comments that the authors may wish to consider.*

We highly appreciate the reviewer's comments. It is true that the manuscript gives a lot of information. Since we do not know what the relevant properties for ice nucleation are, we tried to collect all available information about surface properties and processes. Following the reviewer's suggestions, we rearranged the manuscript and combined results and discussion which are now ordered according to minerals.

*Specific comments*

*I have some thoughts on the interpretation of results concerning the feldspars. Feldspar mineralogy is complex compared to that of the other minerals investigated, is clearly of substantial relevance to the topic in question but is not, to my mind, dealt with thoroughly in this manuscript. Characterization of the feldspar samples is slightly lacking. What is the chemical composition of the three samples used? That the plagioclase sample is named 'andesine' implies a composition but no precise composition is given. The sanidine and microcline samples used could have any composition between 37% K and 100% K from what has been presented. Kauffman et al. (2016) does not contain this information either, as far as I can tell and also does not have an 'andesine' sample.*

The mineralogical composition of sanidine and andesine was determined by Kaufmann et al. (2016) and of microcline by Kumar et al. (2018) using X-ray diffraction (XRD). Rietveld refinement using Profex software (Döbelin and Kleeberg, 2015) was performed for a quantitative analysis. The microcline sample (Si:Al ≈ 3.1; Al:K ≈ 1.4) consists of 86.33% (± 1.71%) microcline, mixed with orthoclase (6.18% ± 0.72%) and albite (7.49% ± 0.48%). The sanidine sample proved to be pure sanidine with Si:Al ≈ 3.1 and Al:K ≈ 1.6 while andesine proved to be pure andesine with Si:Al ≈ 1.7 and Ca/(Ca + Na) = 64%. Note that andesine would be

a labradorite based on Ca/(Ca + Na) ratio but crystallographically it fits best with an andesine. Also note that the andesine sample is termed as "plagioclase" in Kaufmann et al. (2016).

We have added this information in the revised manuscript in Section 2.1 (Lines 99-103).

*I do not think it is reasonable to assume that the ice nucleating ability of the samples used are representative of their crystallographic structures. Harrison et al. (2016) and Whale et al. (2017) observe substantial variability in the ice nucleation effectiveness between crystallographically similar feldspars. Notably, Harrison et al. tested a sanidine sample that nucleated ice with similar effectiveness to most microclines and Whale et al. confirmed that it was sanidine using Raman spectroscopy. The argument is put forward here that the sorts of features that Whale et al. hypothesize cause ice nucleation in larger water droplets are not likely to be present in the smaller droplets used for this study. I am not sure about this. It is quite clear that different alkali feldspars have different structures across multiple scales, including the nanometer scale (Parsons et al., 2015). This is relevant because a) nanoscale features could easily be present on the 'submicrometer' particles used in this paper and b) nanoscale features are most probably on a scale similar to the critical ice nucleus.*

Indeed, the crystallographic structure alone does not determine the ice nucleation ability of mineral dusts, rather, the crystallographic structure determines chemical and physical surface features, which influence the interaction with water and solutes. We agree that nanoscale features are of sufficient size to host a critical ice nucleus. However, even if the sites are small enough to be present on sub-micrometer particles, they might nevertheless be too rare. The type of active sites that are probed in an experimental setup depends on the mineral volume or more precisely the mineral surface present in the sample. For instance, freezing onsets of 251 K and 252 K for microcline correspond to active site densities of about $5 \cdot 10^6$ cm$^{-2}$ and $5 \cdot 10^4$ cm$^{-2}$ for 0.2 wt% and 20 wt% suspension concentrations, respectively (Kumar et al., 2018). These numbers are in excellent agreement with Atkinson et al. (2013) (active site density $10^6$ cm$^{-2}$ for microcline between 251 K and 252 K) who used 14 – 16 μm diameter droplets. Whale et al. (2017) explains the exceptional ice-nucleating ability of alkali feldspars by microtextures related to phase separation into Na and K-rich regions. These sites exhibit densities $< 10^3$ cm$^{-2}$ and are too rare to explain the average freezing on submicrometer particles. The sites inducing ice nucleation above 260 K must be different in some respect to those becoming active only at 250 K or below. The difference might indeed be that the ones freezing at warmer temperature have in addition some suitable perthitic structure.

*It is a little hard to pick out but it appears that the authors think that the plagioclase and sanidine feldspar they have tested are less active than microcline because the sites present on these feldspars are dissolved away effectively immediately on contact with water.*

The dissolution rates that we collected from literature indicate that over the timescales of our emulsion freezing experiments, dissolution leads to relevant surface modifications due to the interaction of water with the feldspar surface. Moreover, we describe the dissolution process, which proceeds at near-neutral conditions via protonation of the oxygen of ≡Al–O–Si≡ bridges with subsequent release of Al$^{3+}$ resulting in an incongruent (deviation of ratio of released Si to Al from stoichiometric ratio) initial dissolution and leading to a ≡Si–OH rich surface. This shows that feldspar surfaces are strongly modified by water. However, it does not tell whether active sites that were initially present on sanidine or andesine are preferentially dissolved away.

*Feldspar dissolution is clearly a complex topic. I do not feel I am qualified to comment on the authors' interpretation of this literature but I would note that it does not appear to be a well settled and understood subject from what I have read. It seems likely that feldspar-to-feldspar variability in dissolution rate is associated with more than just crystal structure and stoichiometry. Still it is helpful that opposing hypotheses for the differing ice nucleation abilities of feldspars now exist, as it is probably straightforward to test which is more consistent with experiments. It would, for instance, be interesting to see if the LD2 sanidine of Harrison et al. and Whale et al. nucleates ice better than a micro-texturally pristine Eifel sanidine (which is what I expect has been tested in this study) in the smaller emulsion droplets.*

Feldspar dissolution is definitely a complex topic, nonetheless, a lot of work has already been done in this field. Given that it is still unclear what really makes feldspars highly IN active (or just any INP in general), it is important to approach the same question by assessing various physicochemical features of surfaces. The longer time scales of emulsion freezing experiments definitely brings surface dissolution in focus, especially when surfaces are exposed to extreme pH conditions.

It would indeed be interesting to see if the LD2 sanidine nucleates ice better than a micro-texturally pristine Eifel sanidine in emulsion freezing experiments. However, the LD2 sanidine could potentially be too coarse (sample sieved through 63 µm mesh (Whale et al., 2017)) to be accommodated by micrometer size droplets.

*The citation for the dissolution rates of feldspars is wrong. I think the authors have read the numbers used from graphs in 'The mechanism of dissolution of the feldspars: Part I. Dissolution at conditions far from equilibrium' but have cited part IV.*

The citation for dissolution rate for microcline and sanidine has been changed to Crundwell (2015) (The mechanism of dissolution of the feldspars: Part I. Dissolution at conditions far from equilibrium, Hydrometallurgy, 151, 151-162, doi:10.1016/j.hydromet.2014.10.006, 2015). Change in citation is made in Table 3 and Section 3.1.4.

**References**

Atkinson, J. D., Murray, B. J., Woodhouse, M. T., Whale, T. F., Baustian, K. J., Carslaw, K. S., Dobbie, S., O/'Sullivan, D., and Malkin, T. L.: The importance of feldspar for ice nucleation by mineral dust in mixed-phase clouds, Nature, 498, 355-358, doi:10.1038/nature12278, 2013.

Crundwell, F. K.: The mechanism of dissolution of the feldspars: Part I. Dissolution at conditions far from equilibrium, Hydrometallurgy, 151, 151-162, doi:10.1016/j.hydromet.2014.10.006, 2015.

Döbelin, N., and Kleeberg, R.: Profex: A graphical user interface for the rietveld refinement program BGMN, Journal of Applied Crystallography, 48, 1573-1580, doi:10.1107/S1600576715014685, 2015.

Kaufmann, L., Marcolli, C., Hofer, J., Pinti, V., Hoyle, C. R., and Peter, T.: Ice nucleation efficiency of natural dust samples in the immersion mode, Atmos. Chem. Phys., 16, 11177-11206, doi:10.5194/acp-16-11177-2016, 2016.

Kumar, A., Marcolli, C., Luo, B., and Peter, T.: Ice nucleation activity of silicates and aluminosilicates in pure water and aqueous solutions – part 1: The K-feldspar microcline, Atmos. Chem. Phys., 18, 7057-7079, doi:10.5194/acp-18-7057-2018, 2018.

Whale, T. F., Holden, M. A., Kulak, A. N., Kim, Y.-Y., Meldrum, F. C., Christenson, H. K., and Murray, B. J.: The role of phase separation and related topography in the exceptional ice-nucleating ability of alkali feldspars, Phys. Chem. Chem. Phys., 19, 31186-31193, doi:10.1039/C7CP04898J, 2017.

---

## Author Response (AR2)

We give below our responses (in black) to Co-Editor's comments (*in blue*)

*Thank you for submitting your revised manuscript. I especially appreciate your efforts to meet the referees' requests to shorten and re-organize the paper. I recognize the value in the length of the discussion of this and numerous prior related studies to build a comprehensive understanding of the factors that control the ice nucleation ability of mineral particles and surfaces. This series of three papers, and this third paper in particular with its summary and discussion of the main results from all three papers, will provide very valuable information to the ice nucleation and atmospheric sciences community, and the geosciences.*

We thank the co-Editor for the appreciation and constructive comments.

*I repeat the same suggestions below as for the Part 2 paper regarding closely related prior studies on the effects of chemical aging on the ice nucleation properties of mineral dust particles. It is up to you if you wish to add a discussion of these papers to both part 2 and part 3, or just in one of these papers.*

*In the Introduction, and elsewhere, there are some other well-cited studies of the effects of chemical aging on the ice nucleation properties of mineral dust that you might consider including and discussing briefly. The FROST-2 campaign at LACIS condensed sulphuric acid onto mineral aerosol particles, and found this inhibited the deposition freezing, while immersion freezing was partially decreased.*

*Sullivan, R. C.; Petters, M. D.; DeMott, P. J.; Kreidenweis, S. M.; Wex, H.; Niedermeier, D.; Hartmann, S.; Clauss, T.; Stratmann, F.; Reitz, P.; Schneider, J.; Sierau, B. Irreversible loss of ice nucleation active sites in mineral dust particles caused by sulphuric acid condensation. Atmos. Chem. Phys. 2010, 10, 11471–11487.*

*Reitz, P.; Spindler, C.; Mentel, T. F.; Poulain, L.; Wex, H.; Mildenberger, K.; Niedermeier, D.; Hartmann, S.; Clauss, T.; Stratmann, F.; Sullivan, R. C.; DeMott, P. J.; Petters, M. D.; Sierau, B.; Schneider, J. Surface modification of mineral dust particles by sulphuric acid processing: implications for ice nucleation abilities. Atmos. Chem. Phys. 2011, 11, 7839–7858, doi:10.5194/acp-11-7839-2011.*

*Niedermeier, D.; Hartmann, S.; Wex, H.; Clauss, T.; Kiselev, A.; Sullivan, R. C.; DeMott, P. J.; Petters, M. D.; Reitz, P.; Schneider, J.; Mikhailov, E.; Reimann, B.; Bundke, U.; Stetzer, O.; Sierau, B.; Shaw, R. A.; Mentel, T. F.; Stratmann, F. Experimental study of the role of physicochemical surface processing on the IN ability of mineral dust particles. Atmos. Chem. Phys. 2011, 11, 11131–11144.*

The suggested references have been added to the revised manuscript (lines 62 – 64).

*In related experiments, Arizona test dust (which Atkinson et al. later found contains K-feldspars, which explains the high INA of this dust) was exposed to nitric acid vapor. This was found to impair deposition freezing while immersion freezing was unaltered. The possible reasons for the differences between the effects of sulphuric and nitric acid were discussed. It was concluded that the nitric acid reaction products dissolve off the surface in the immersion mode to either re-expose the original ice active sites, or new surface sites if irreversible chemical alteration of the surface resulted.*

*Sullivan, R. C.; Miñambres, L.; DeMott, P. J.; Prenni, A. J.; Carrico, C. M.; Levin, E. J. T.; Kreidenweis, S. M. Chemical processing does not always impair heterogeneous ice nucleation of mineral dust particles. Geophys. Res. Lett. 2010, 37, doi:10.1029/2010GL045540.*

The suggested reference has been added to the revised manuscript (lines 64 – 66).

*The role of the reaction product versus chemical alteration of surface sites was then investigated by Sihvonen et al.:*

*Sihvonen, S. K.; Schill, G. P.; Lyktey, N. A.; Veghte, D. P.; Tolbert, M. A.; Freedman, M. A. Chemical and physical transformations of aluminosilicate clay minerals due to Acid treatment and consequences for heterogeneous ice nucleation. J. Phys. Chem. A 2014, 118, 8787–96, doi:10.1021/jp504846g.*

The suggested reference has been added to the revised manuscript (lines 66 – 68).

*Finally, just recently a study on the effect of secondary organic aerosol coatings on the INA of mineral dust concluded that there was no discernible effect in the immersion freezing mode. Similar to the nitric acid aging, presumably the SOA is dissolved off the surface when immersed in droplets to uncover the surface sites. This paper was just recently accepted for publication in ACP:*

*Heterogeneous Ice Nucleation Properties of Natural Desert Dust Particles Coated with a Surrogate of Secondary Organic Aerosol*

*Zamin A. Kanji, Ryan C. Sullivan, Monika Niemand, Paul J. DeMott, Anthony J. Prenni, Cedric Chou, Harald Saathoff, and Ottmar Möhler*

*Atmos. Chem. Phys. Discuss., https://doi.org/10.5194/acp-2018-905, 2018*

*Revised manuscript accepted for ACP (discussion: closed, 4 comments)*

The suggested reference has been added to the revised manuscript (lines 68 – 70).

[revised manuscript text omitted]